# Rising Temperatures, Falling Leaves: Predicting the Fate of Cyprus’s Endemic Oak under Climate and Land Use Change

**DOI:** 10.3390/plants13081109

**Published:** 2024-04-16

**Authors:** Konstantinos Kougioumoutzis, Ioannis Constantinou, Maria Panitsa

**Affiliations:** Laboratory of Botany, Department of Biology, University of Patras, 26504 Patras, Greece; kkougiou@aua.gr (K.K.); yiannis_constantinou@hotmail.com (I.C.)

**Keywords:** biodiversity conservation, extinction risk, Golden Oak, LULC, Mediterranean

## Abstract

Endemic island species face heightened extinction risk from climate-driven shifts, yet standard models often underestimate threat levels for those like *Quercus alnifolia*, an iconic Cypriot oak with pre-adaptations to aridity. Through species distribution modelling, we investigated the potential shifts in its distribution under future climate and land-use change scenarios. Our approach uniquely combines dispersal constraints, detailed soil characteristics, hydrological factors, and anticipated soil erosion data, offering a comprehensive assessment of environmental suitability. We quantified the species’ sensitivity, exposure, and vulnerability to projected changes, conducting a preliminary IUCN extinction risk assessment according to Criteria A and B. Our projections uniformly predict range reductions, with a median decrease of 67.8% by the 2070s under the most extreme scenarios. Additionally, our research indicates *Quercus alnifolia*’s resilience to diverse erosion conditions and preference for relatively dry climates within a specific annual temperature range. The preliminary IUCN risk assessment designates *Quercus alnifolia* as Critically Endangered in the future, highlighting the need for focused conservation efforts. Climate and land-use changes are critical threats to the species’ survival, emphasising the importance of comprehensive modelling techniques and the urgent requirement for dedicated conservation measures to safeguard this iconic species.

## 1. Introduction

Climate and land-use alterations will significantly threaten plant diversity in the ensuing decades [1], markedly influencing species abundances, phenologies, and community composition, with expectations of escalating impacts [2], potentially leading to the loss of up to 31% of known plant species [3]. The Mediterranean Basin, recognised both as a global biodiversity hotspot [4] and a climate change hotspot [5,6], epitomises these challenges. It has already manifested ecological shifts, including phenological alterations [7], reduced provision of regulating ecosystem services [8], and species loss [9,10,11,12,13,14], with these alterations predicted to intensify, especially in Mediterranean forests [8,15,16]. Conversely, land-use modification has emerged as a primary factor in plant extinctions over recent centuries [17], significantly transforming continental Europe’s vegetation [18] and causing excessive soil erosion in Mediterranean forests [5]. These phenomena present formidable challenges for Mediterranean islands, necessitating a profound comprehension of ecological impacts to devise efficacious conservation strategies [2].

As the Mediterranean Basin, a region integral to global biodiversity, confronts these unprecedented ecological shifts, our study aims to contribute to this understanding by focusing on Cyprus and, more specifically, to address critical knowledge gaps surrounding climate and land-use change impacts on *Quercus alnifolia*, an endemic keystone oak with limited dispersal capacity, facing unique threats within the Mediterranean Basin. Noted as the third largest island in the Mediterranean [19] and a regional (sensu [4]) hotspot [20,21] within the broader Mediterranean Basin hotspot, Cyprus embodies these broader challenges. The island currently enjoys a temperate climate characterised by warm, dry summers, although certain areas are deemed hot and arid according to the Köppen–Geiger classification [6]. However, Cyprus is considered a climate change hotspot [22,23], as projections indicate a significant increase in arid and dry zones across Cyprus in the forthcoming years, with only a minor portion of the island expected to maintain a typical Mediterranean climate [6]. Cyprus is one of the most isolated islands in the eastern Mediterranean [24] and hosts a diverse flora [24,25], attributed to its distinct geographic and climatic conditions, prolonged isolation, oceanic origin, and long history of human presence [21,24,25]. The flora of Cyprus comprises 1649 native taxa [26], with an endemism rate of 8.85% (146 endemics; [26]), aligning with the rates observed in other Mediterranean islands [27]. The IUCN Red List categorises 298 vascular plant species from Cyprus, with the majority listed as Least Concern (254 taxa; 85.2%) and only 22 as threatened (7 as Critically Endangered, 5 as Threatened and 10 as Vulnerable; [25,28]). Notably, certain zones within the broader Mt. Troodos region are recognised as critical for the richness of threatened species [28], as Mt. Troodos hosts half of the native plants occurring in Cyprus and 70% of the Cypriot endemics [29]. Despite its rich biodiversity and well-established network of protected areas [30], Cyprus grapples with considerable anthropogenic threats [31], such as land degradation [32], echoing the broader challenges faced by the Mediterranean Basin. Cyprus thus offers a distinctive context for examining the broader implications of climate and land-use changes in Mediterranean ecosystems.

Undertaking climate change vulnerability assessments is recommended for rare, threatened, and endemic taxa, according to the IUCN [33,34]. Extensive research has been conducted on the effects of climate change on Mediterranean ecosystems (e.g., [13,16]). Forests span approximately 9.4% of the world’s total forest area and account for nearly 30% of the Mediterranean’s land surface [35]. These forests are already experiencing the impacts of climate change [36], potentially paving the way for the rise of adverse conditions. Such conditions could ultimately undermine their resilience, diminish their adaptability and survival prospects, and curtail their spatial coverage (e.g., [13,14]). Regarding Mediterranean trees, research has predominantly concentrated on the representatives of the genus *Quercus*, with a relatively wide Mediterranean distribution [13,37,38]. This emphasis is likely due to *Quercus* being among the richest genera within the tree families [39], coupled with its significant medicinal, ecological, and economic attributes [40]. Additionally, forest ecosystems predominantly comprising oak species are increasingly recognised for their multifaceted utility, encompassing the provision of ecosystem services, the generation of biomass for bioenergy, and the production of wood fuel [38,41,42]. They are also valued for yielding high-quality timber for industrial use and offering vital habitats for diverse wildlife species [38,41,42]. Climate change is one of the major threats oaks will face in the future, despite having survived several unfavourable periods since their emergence 55 Mya [43,44]. Research has also probed into other genera, including *Pinus* [45], *Juniperus* [11,14], *Abies* [14,46,47], and *Cedrus* [46,48], with the primary emphasis being on the western Mediterranean and the eastern Mediterranean countries, encompassing Asia Minor and the Levantine coast (i.e., Turkey, Israel, Lebanon, and Egypt). Climate change thus threatens the habitat suitability, growth, regeneration capacity, and distribution ranges of many Mediterranean tree species [49], and increased drought stress and extreme heat are expected to cause tree growth declines, higher mortality risks, and increasing disturbance from fires, pests, and diseases [50]. The impacts of climate change may exceed the resilience and adaptive capacity of many relict Mediterranean tree populations that have persisted in microrefugia, thus causing local extinctions [51,52].

Despite hosting a highly diverse single island endemic flora, second only in the Eastern Mediterranean to Crete [21], another island biodiversity hotspot faced with an elevated extinction risk over the coming decades [53,54], research on the impacts of climate and land-use change on its endemic plant taxa has been extremely limited [55,56], particularly for Cypriot endemic species such as *Quercus alnifolia* Poech [57,58,59,60]. 

*Quercus alnifolia* is an evergreen shrub or tree, endemic to the Troodos mountain range [61,62], occupying an area spanning approximately 23,700 hectares [63]. This species, which can reach up to 14 m in height, predominantly forms vast forests on ultramafic soils, thriving at altitudes ranging from 400 to 1800 m above sea level [64]. Within the lower reaches of its altitude spectrum, *Quercus alnifolia* is typically found in moister sites, with its presence gradually diminishing in favour of taxa that demonstrate greater drought tolerance. Conversely, at the higher end of its altitudinal habitat, the species endures comparatively severe winters, with temperatures plummeting to as low as −15 °C and snow persisting for three to four months annually [63,65]. Being a keystone species, it holds a central function in maintaining the ecological stability of Cyprus [61,62]. The loss of this species due to climate and land-use changes could disrupt trophic relationships and water cycles within the Troodos mountain range, leading to a cascade of ecological consequences [61,62]. It plays a critical role in soil stabilisation and post-disturbance regeneration, as it regenerates after fires or felling and serves as a food source for local fauna [61,62]. Despite its adaptability in arid habitats [61,62], as it creates wide-ranging life-supporting environments where other trees struggle to survive [66], it faces threats from fires, logging, grazing, and agricultural, infrastructural, and urban development [42,67,68] and displays low genetic diversity due to genetic drift [69,70,71], as do many other narrow Mediterranean endemics [72]. *Quercus alnifolia* is listed as Least Concern (LC) in the IUCN Red List and Vulnerable (VU) in the Red List of Oaks [42,67,68]. Protected under Habitat Directive 92/43/EEC and forest law, scrub and low forest vegetation with *Quercus alnifolia* is recognised as a priority habitat type (9390*) in Annex I [73]. Moreover, a Plant Micro-Reserve was established in the Natura area, “Koilada Kedron—Kampos”, to promote the conservation status of this priority habitat type [74], as for other Cypriot endemics. Despite these protective measures and its recognition as Cyprus’s national tree [66], there is a distinct absence of research investigating the influence of climate and land-use changes on *Quercus alnifolia*. Its IUCN report [67] explicitly indicates that climate change is expected to significantly diminish the species’ Extent of Occupancy (EOO) and Area of Occupancy (AOO). Consequently, monitoring of both the species’ population and habitat to fully understand climate change’s impact has been advised [67]. 

This creates a pressing need for research that not only broadens our understanding of how endemic species respond to environmental stressors but also aids in the development of targeted conservation strategies. In light of this, we aim to provide a rigorous assessment of the potential impacts of environmental changes on *Quercus alnifolia*, an emblematic plant species of the Cypriot flora. Standard Species Distribution Models (SDMs) hold immense potential for ecological insight. However, when applied to species with limited dispersal potential, they often overestimate future habitat expansion possibilities [75,76]. Projecting realistic tree migrations necessitates integrating dispersal characteristics. Seed size and the type of dispersal vector (wind, fauna) play a crucial role in determining species movement potential [77]. This integration with dispersal limits becomes all the more urgent when set against the pace of projected climate change. Over the next century, many tree species may need to migrate across hundreds of kilometres to remain within climatically suitable areas [78]. In contrast to past climate fluctuations, projected climate velocity could far exceed even the maximum migration of species traditionally assumed to have greater dispersal potential [79]. This study directly confronts this challenge by focusing on *Quercus alnifolia*, an endemic tree with large, heavy seeds (up to 3.6 g) and an estimated maximum dispersal distance of 1500 m [80]. While this work goes beyond traditional SDMs by acknowledging *Quercus alnifolia*’s limited dispersal, a purely climate-focused SDM is only the first step. Understanding how soil parameters will further constrain the suitability of seemingly ideal habitats requires careful selection of abiotic variables [81]. Ignoring projections across less readily available soil and hydrological dimensions may result in the appearance of vast, suitable habitats that are inaccessible due to dispersal barriers and changing soil conditions. This highlights the potential for overstating model results when static variables’ apparent importance masks their inability to represent real-world complexity and the species’ ecological needs [82,83]. Realistic conservation goals can be established only by explicitly integrating dispersal dynamics and a more comprehensive array of abiotic factors tailored to the species’ ecology. To this end, we utilised a Species Distribution Modelling framework to delineate the present and prospective distribution of *Quercus alnifolia*, incorporating a suite of abiotic variables known to enhance the predictive power and robustness of environmental niche models in plants. This included a focus on soil variables [84], crucial for understanding plant species distributions [85], alongside a comprehensive array of environmental variables from ENVIREM [86], bioclimatic data from WorldClim [87,88,89] and land-use data [90]. In a novel approach, we also incorporated both current and projected soil erosion data [91] into our models, as well as key hydrological variables [92], in an attempt to cover a large portion of the species’ niche [93,94,95], contrary to standard practices [94,95,96,97]. While climate and land-use changes are acknowledged drivers of habitat alteration, the potential additive effect of hydrological variables and future soil erosion represents a significant, yet underexplored, facet of ecological research. By incorporating these projections, our study not only addresses a critical gap in current ecological models but also offers a more comprehensive understanding of the multifaceted pressures on *Quercus alnifolia*, a species intrinsically linked with soil stabilisation in the Troodos mountain range, laying the groundwork for targeted conservation efforts amidst rapidly changing environmental conditions. Our research builds upon previous studies in Greece [10,11,12] integrating land use and land cover (LULC) data into climate change vulnerability assessments (CCVAs). It is the first in Cyprus and the eastern Mediterranean to explicitly combine LULC with hydrological and soil erosion data, as recommended by [98]. CCVAs excluding current and projected LULC data could lead to inaccurate assessments of future extinction risk [98]. The integration of hydrological variables and future soil erosion data alongside climate and land-use change projections into our SDMs paves the way for a series of pertinent research questions. These questions aim to dissect the intricate and complex dynamics of species distribution, vulnerability, and resilience, offering a nuanced and detailed understanding of how *Quercus alnifolia* may navigate the impending ecological transformations. In conclusion, we aim to:(a)map the future potential distribution of *Quercus alnifolia* under climate and land-use change scenarios,(b)critically evaluate *Quercus alnifolia*’s sensitivity, exposure, and vulnerability to these changes, and(c)conduct a preliminary IUCN extinction risk assessment at a global scale under Criteria A and B, thereby offering actionable insights for conservation strategies and policy-making.

The comprehensive insights gleaned from our study hold the potential to significantly influence policy and conservation strategies, not only for *Quercus alnifolia* but also for similar species within the Mediterranean and other biodiversity hotspots. By mapping potential distributions under future scenarios, evaluating species sensitivity and vulnerability, and conducting an extinction risk assessment, our research offers a robust foundation for informed decision-making. This work underscores the urgent need for integrating scientific insights into policy frameworks, aiming to mitigate the impacts of climate and land-use changes through adaptive management and conservation planning. In doing so, we contribute to biodiversity conservation, ensuring the resilience and sustainability of Mediterranean ecosystems in the face of unprecedented environmental challenges. 

## 2. Results

### 2.1. Land Use and Land Cover Changes

Conifers dominate the current landscape of the study area. Broad-leaf evergreen temperate shrubs hold the second-largest extent, followed by croplands, barren areas, and broad-leaf deciduous temperate shrubs (Figure 1). Model projections for the study area show increased coverage by conifers throughout the century (Figure 2 and Appendix A). At the same time, the projections show an expansion in the area of abandoned cropland (Figure 2 and Appendix A). These projections also show croplands transitioning to broadleaf deciduous temperate trees and evergreen shrubs, with a higher relative loss rate in these areas than other LULC classes (Figure 2 and Appendix A). Most of the study area shows stability in LULC classifications, with 1–4 LULC transition steps primarily occurring in the southern regions of the Troodos mountain range (Figure 3 and Appendix A).

### 2.2. Species Distribution Models

The model demonstrated robust performance (Appendix A), significantly exceeding random expectations (*p* < 0.01). *Quercus alnifolia* presented minimal potential niche truncation (PTNI = 0.05). Among all the response variables, the proportion of clay particles in the fine earth fraction, slope, and field capacity had the highest contribution to explaining *Quercus alnifolia* distribution (Appendix A). Moreover, among the temporally dynamic variables, barren areas, Thornthwaite’s aridity index, and coverage by conifers are the key contributors to the spatial distribution of *Quercus alnifolia* (Appendix A). Given that the LULC variables related to *Quercus alnifolia* show little variation over time (see Section 2.1. Land Use and Land Cover Changes), we focus on how the other temporally dynamic variables might impact the species’ future distribution. While the statistical significance of soil erosivity is lower than other variables, *Quercus alnifolia* is present across various soil erosion conditions, persisting in areas with moderate to high soil erosion (Figure 4). Additionally, *Quercus alnifolia* occurs in relatively arid climates with a narrow annual temperature range (approximately 28–29 °C; Figure 4) and high temperature seasonality (approximately 6.5–7 °C; Figure 4). *Quercus alnifolia* is found in areas with a relatively high minimum temperature during the warmest quarter (around 18 °C; Figure 4) and low precipitation during the driest month (approximately 4–6 mm; Figure 4). We primarily concentrate on the HadGEM2 GCM RCP 8.5 SSP5 scenario for the 2070s, highlighting it as the scenario projecting the most pronounced alterations in the species’ range (Appendix A). 

### 2.3. Habitat Suitability Range Change

In the study area, we did not detect univariate extrapolation (i.e., extrapolation outside the range of training conditions). However, we observed combinatorial extrapolation (i.e., extrapolation within the range of training conditions), predominantly in the eastern and southern parts of the species’ range (Figure 5). Consequently, we assigned suitability values of 0 to those areas. Regarding the Shape extrapolation threshold, after evaluating seven different thresholds ranging from 50 to 300 in increments of 50, we selected a threshold of 100. This decision was informed by the relatively broad distribution of *Quercus alnifolia* within the Troodos mountain range, notwithstanding its status as a single-island endemic. 

*Quercus alnifolia* is predominantly found in the principal Troodos mountain range, with the central part of its distribution located in Mt. Troodos National Park (Figure 6A and Figure 7A). Future projections suggest a more fragmented and diminished distribution for the species (Figure 6B and Figure 7B).

*Quercus alnifolia* is anticipated to experience range reductions (Figure 8), with these contractions intensifying over time. The overall median range reduction is estimated at −63.8%, with projections for the 2050s and the 2070s at −55.2% and −67.8%, respectively (Appendix A). 

*Quercus alnifolia* is likely to experience an altitudinal shift towards statistically significantly lower altitudes in the future (mean current altitude: 1160 m a.s.l.; Kruskal–Wallis: H = 1881.6, d.f. = 24, *p* < 0.001), particularly under the most pessimistic Global Circulation Model (GCM), HadGEM2 (minimum mean future altitude: 1005 m a.s.l.; Appendix A; Figure 6). Conversely, a shift to statistically significantly higher altitudes (maximum mean future altitude: 1322 m a.s.l.; Kruskal–Wallis: H = 1881.6, d.f. = 24, *p* < 0.001) is expected under the most optimistic GCM, CCSM4 (Appendix A; Figure 9). The divergence in these outcomes, driven by HadGEM2 and CCSM4, highlights the inherent uncertainty associated with GCM-derived climate projections, particularly in topographically complex regions like the Troodos mountain range, and may be partly attributable to differences in how HadGEM2 and CCSM4 represent orographic precipitation and temperature gradients. All fragmentation metrics, except for the number of patches (in eight cases regarding the 2070s), have lower future values than the baseline (Appendix A). Specifically, the patch cohesion index decreases and fragmentation increases over time (Appendix A). Based on the significant reduction in effective mesh size, the patches are becoming smaller and more numerous (Appendix A). Furthermore, based on the decrease in the mean and maximum patch areas and their standard deviations, the patches are shrinking and varying more in size (Appendix A). The landscape is splitting into more, smaller, and isolated patches during the 2070s, as the patch number increases during that period (Appendix A).

### 2.4. Sensitivity, Exposure, and Vulnerability to Climate and Land-Use Change

*Quercus alnifolia*’s sensitivity was classified as moderate (Appendix A). In a temporal context, exposure steadily increased, peaking in the 2070s for the HadGEM2 RCP 8.5 SSP5 combination (Appendix A). Notably, the HadGEM2 GCM demonstrated a significantly higher median exposure value (92.90 ± 1.81) compared to the CCSM4 GCM (65.4 ± 10.9; Kruskal–Wallis: H = 17.325, d.f. = 1, *p* < 0.001; Appendix A). A similar pattern was observed in the vulnerability assessment, where the HadGEM2 GCM showed significantly greater median vulnerability (1.70 ± 0.03) compared to the CCSM4 GCM (1.56 ± 0.05; Kruskal–Wallis: H = 17.325, d.f. = 1, *p* < 0.001; Appendix A).

When mapping the spatial distribution of vulnerability, our analysis concentrated on the HadGEM2 RCP 8.5 scenario for the 2070s, which was identified as the most severe scenario in terms of vulnerability. Regions of heightened vulnerability were primarily located on Mt. Troodos and along the western edge of the species’ distribution range (Figure 10).

### 2.5. IUCN Extinction Risk Assessment

The preliminary extinction risk assessment, employing IUCN Criteria A and B individually and in combination, categorises *Quercus alnifolia* for the baseline period and all future periods as:Critically Endangered, according to the IUCN Criterion A assessment,Least Concern or Near Threatened in most scenarios, with the exception of the CCSM4 4.5 GCM/RCP combination in the 2070s and the CCSM4 4.5 SSP3 GCM/RCP/SSP combination in the 2050s (Vulnerable), according to the IUCN Criterion B assessment andCritically Endangered, when considering the combined IUCN Criteria A and B assessment.

Detailed results encompassing all scenarios are presented in Appendix A.

## 3. Discussion

Despite its recognition as Cyprus’ national tree and its ecological significance, there has been a lack of research on how climate and land-use changes affect the distribution of *Quercus alnifolia*. Addressing this gap, we responded to the call by [67] and conducted a detailed assessment of its vulnerability to climate and land-use change. This assessment incorporated previously overlooked factors in Species Distribution Models and included dispersal constraints to predict the species’ niche and potential range shifts more accurately. *Quercus alnifolia* is projected to face altitudinal displacement, severe fragmentation, and significant range reductions of up to −70.3%, with these intensifying over time. According to IUCN Criteria A and B, the species’ risk of extinction is set to increase in the forthcoming decades, making it particularly susceptible to alterations in its core habitat due to environmental changes.

### 3.1. Abiotic Variables Affecting the Distribution of Quercus alnifolia

Ecological processes, operating across multiple levels of complexity, drive species distributions and manifest across diverse spatial scales. Abiotic variables can thus be distinguished between regional drivers [99], where climatic factors dominate plant distribution and abundance patterns [100], and local-scale determinants [99], such as topography and soil conditions [101]. Integrating a range of environmental variables thus offers a robust, multi-scale framework for predictive species distribution modelling, capturing a more holistic understanding of species’ dynamics.

We have specifically integrated temporally static hydrological variables and temporally dynamic soil erosion data, thus offering a more comprehensive depiction of the species’ niche [93,96], contrary to standard practices [94,95,96,97], as similar studies sometimes rely solely on readily accessible environmental data, failing to incorporate ecologically meaningful variables that directly influence species distributions [81]. These omissions frequently lead to inadequate modeling outcomes that disregard crucial ecological drivers and thus risk incomplete or erroneous estimations of range dynamics [81]. The overrepresentation of temporally static variables in SDMs may lead to inflated variable importance and may also lead to potentially misleading inferences regarding habitat suitability [82,83]. This does not mean that topography and aspects of soil composition are not significant for *Quercus alnifolia*’s distribution (Appendix A); on the contrary, including these factors leads to higher model precision and accuracy, aligning with other studies on plant species distribution modelling [85,96,102]. Nevertheless, one must interpret their apparent predictive power cautiously, as static variables in SDMs can exhibit inflated importance. A critical concern when employing temporally stable variables for predictive modeling is reduced model transferability in space and time. However, our rigorous assessment of potential extrapolation (Figure 5), combined with evidence of minimal niche truncation (PTNI = 0.05), supports the robustness and predictive credibility of our findings. This methodological approach indicates reduced uncertainties associated with both extrapolation and niche truncation limitations [103,104]—ultimately boosting model transferability.

Dynamic variables such as land-use cover, albeit exhibiting low temporal variability within the study area (Figure 2, Figure 3 and Appendix A), are ecologically crucial factors affecting habitat suitability and model accuracy and precision in *Quercus alnifolia* (Appendix A) and other oaks and endemic or range-restricted plants [10,11,12,41]. While our models revealed greater sensitivity to land-use variables than climatic variables, current forecasts indicate minimal anticipated land-use change within *Quercus alnifolia*’s range (Figure 2, Figure 3 and Appendix A). However, land degradation and land development pressures related to tourism [2,31,32] can have detrimental impacts on endangered species [105] and thus warrant careful monitoring of any changes adjacent to protected habitat reserves where *Quercus alnifolia* occurs. This focus becomes vital given the projected high sensitivity of this endemic species to even subtle shifts in land-use patterns (Appendix A).

These considerations withstanding and given the absence of temporal variation in pedological and topographical variables and the little temporal variation in the LULC variables in the study area (Figure 2, Figure 3 and Appendix A), our results indicate that *Quercus alnifolia* currently tolerates and will continue to tolerate moderate to high soil erosion conditions (Figure 4), potentially being more adapted to tolerating variations in future erosion trends. Thus, *Quercus alnifolia* will likely remain a vital keystone species in Cypriot mountain ecosystems, safeguarding against erosion, optimising nutrient cycles, ensuring forest health and biodiversity [61,62], and providing crucial ecosystem services [29].

The distribution of tree species predominantly relies on climatic factors [106], at least at regional scales [100], with temperature and water availability, including precipitation, playing critical roles [107]. Temperature-related variables are among the main drivers of oak species’ distribution [13,38]. *Quercus alnifolia* seems to thrive within a narrow annual temperature range, with high temperature seasonality and a relatively high minimum temperature during the warmest quarter (Figure 4). Temperature seasonality plays a central role in shaping the distribution of oaks [108] and reflects adaptations to alpine and sub-alpine environments [109]. Oaks of Section *Ilex*, to which *Quercus alnifolia* belongs, carry functional leaf traits suggesting ancestral pre-adaptation to seasonal drought periods and thus indicating evolutionary adaptation to semi-arid environments [110]. This adaptability, however, might become decoupled due to *Quercus alnifolia*’s low genetic diversity [69,70,71], which in turn may hinder its response to increasingly severe shifts in seasonal climate patterns.

The importance of Thornthwaite’s aridity index, precipitation seasonality, and precipitation of the driest month is related to the biological cycle of *Quercus alnifolia*, as its flowering season lasts from April to May [26]. Precipitation seasonality, particularly in terms of drought duration and intensity, is a vital climatic variable for oak species [108]; these dry periods directly constrain oak growth due to reduced photosynthetic rates and potentially increase mortality risks [111]. Despite *Quercus alnifolia*’s drought resistance and adaptability to arid climates, severe water stress due to limited soil moisture may lead to elevated seed mortality and reduced seed germination if acorn desiccation drops below 35% moisture content [73]. Climate change projections for Cyprus warn of increasing water scarcity caused by changing precipitation patterns, heightened evapotranspiration, and intensified rainfall events [22,23]. These disruptions could adversely impact seed maturation and dispersal timings in *Quercus alnifolia*—likely desynchronizing vital processes from favourable germination conditions and potentially triggering even greater range contractions. This concern aligns with our model, which demonstrates the importance of field capacity for the species’ distribution (Appendix A). Seed drying sensitivity may affect propagation success in several species, and reduced precipitation may negatively impact seed germination success [112] and production [113]. Consequently, this may lead to drought-adapted *Quercus* species being gradually replaced by other tree species, such as pines, better suited to summer droughts [114], as projected in other Eastern Mediterranean countries [14].

### 3.2. Habitat Suitability Range Change and IUCN Extinction Risk Assessment

The Mediterranean region is considered highly vulnerable to climate change impacts [5], like increasing temperatures, more frequent heatwaves, decreasing precipitation, increasing aridity, and altered fire regimes. This is expected to negatively impact Mediterranean trees [115], decreasing suitability in established ranges and leading to die-offs or reduced forest health. The increasing frequency and intensity of fires is one of the most serious impacts and threats to forest ecosystems, mainly to the forest endemic taxa of southern and insular areas [116]. Mediterranean trees inhabiting mountainous regions face considerable risk of habitat loss under shifting climate regimes [16,45]. This potential reduction is particularly acute for range-restricted and endemic species [11,13] and is especially true for one of the largest tree genera [39], *Quercus*, as climate change is one of the major threats oaks will face in the future [42,44]. Forecasts indicate a contraction in the habitats of certain oak species across the Mediterranean [13,14,38,108]. Conversely, species such as *Quercus petraea* and *Quercus pubescens* are projected to witness an expansion of their distribution ranges, extending from the Mediterranean basin into Central Europe [117]; the latter, however, is projected to face range declines, at least locally, in Greece and in Italy [14,118]. *Quercus alnifolia* is no exception to this general rule, as our results indicate (Appendix A; Figure 6, Figure 7 and Figure 8).

Species distribution models that rely solely on climatic variables might misrepresent the true extent of projected range shifts for sensitive species, leading to either risk overestimations or underestimations and incomplete quantification of the species’ niche [55,95,119]. While Mediterranean oaks, cedars, firs, and pines face significant pressures from climate change, most of these studies only focus on standard bioclimatic variables, or some incorporate temporally static soil or topographical variables [13,14,45,46,51]. Moreover, analyses omitting realistic dispersal scenarios—and other site-specific soil or hydrological constraints, as well as the incorporation of land-use change data—may underestimate potential contractions in suitable habitat, as they ignore the critical influence of dispersal constraints and land-use modifications on range shifts [119,120,121] and may lead to lower model performance [81]. We explored how model refinement may alter outcomes by conducting a sensitivity analysis (Appendix A). Incorporating realistic dispersal constraints and variables ranging from climate-only to combinations including topography, soil, hydrology, and land-use data revealed the substantial extent to which simplified models may misrepresent environmental limitations for *Quercus alnifolia*. Results illuminate the significant gap between potential *Quercus alnifolia* range loss estimated only with climate variables without realistic dispersal constraints versus projections integrating multiple abiotic factors and dispersal limitations (+174.5% and −70.3%, respectively; Appendix A; Figure 6, Figure 7 and Figure 8). In species distribution models, two contrasting dispersal scenarios are typically considered: unlimited dispersal, where a species can occupy any suitable area, and no dispersal, where it cannot follow its preferred environmental conditions [121], though the reality is often somewhere in between [122]. The choice between these scenarios, often made due to the simplicity of application or lack of detailed ecological data (e.g., lifecycle, seed characteristics, dispersal vectors, and methods) [123], is usually based on fundamental species characteristics such as the level of endemicity and life form. Without incorporating dispersal limits and a comprehensive set of environmental variables that closely mirror the species’ niche into our analysis, *Quercus alnifolia* might be expected to thrive in a warmer and drier world. This expectation could appear justified by its ability to adapt to arid conditions, fostering extensive habitats supportive of life where other species falter, and its resilience in regenerating after disturbances like fires [61,62,66]. However, the impending climate and land-use changes pose a significant threat to the capacity of *Quercus alnifolia* to maintain its niche, with projections indicating substantial range reductions that become increasingly severe with future projections (Appendix A; Figure 8). The projected trend indicates severe habitat loss accompanied by fragmentation, leading to smaller, more numerous, and isolated patches that vary in size (Appendix A; Figure 6 and Figure 8). Fragmentation impedes plant migration through several mechanisms: decreased habitat and seed establishment potential, compromised gene flow vital for adaptability, and reduced propagule pressure [79,124,125]. Thus, increased fragmentation may apply further pressure on the species’ survival.

*Quercus alnifolia* may undergo upward or downward altitudinal shifts depending on the Global Circulation Model (GCM) utilised, with the absolute mean displacement reaching 155 m (Appendix A; Figure 9). The divergence in these outcomes, driven by HadGEM2 and CCSM4, highlights the inherent uncertainty associated with GCM-derived climate projections, particularly in topographically complex regions like the Troodos mountain range, and may be partly attributable to differences in how HadGEM2 and CCSM4 represent orographic precipitation and temperature gradients. While these directions differ, significant elevational displacement appears to be a probable risk with potentially complex ramifications for this tree’s adaptation. These shifts could instigate interactions with other taxa, possibly disrupting established community dynamics. Altitudinal displacement has also been observed in or predicted for other Mediterranean trees and endemic or range-restricted species [15,53,108,126,127,128,129,130,131], which may lead to the reshuffling of high-altitude forest composition and a change in dominant trees, with alpine conifers potentially substituted by oaks and Mediterranean pines [132,133]. This phenomenon, in turn, may alter the provision of ecosystem services in high mountain forests [134]. Oaks may also be outcompeted for newly suitable areas of expansion by faster-dispersing competitors like Aleppo Pine [114]. This phenomenon might also be observed in Cyprus [135], as increasing heat and drought render former occurrence areas of *Quercus alnifolia* less hospitable. This underscores the severity of climate- and land-use-driven range reductions, fragmentation, and competitive pressures predicted for this iconic Cypriot endemic tree. Our model outcomes mirror those for *Abies marocana* [51] and *Cedrus libani* [136], other characteristic Mediterranean trees facing overwhelming range contraction over the coming decades, and indicate the stark possibility that without proactive climate-focused conservation, such emblematic Mediterranean trees could dwindle towards local extinction—a grave cost for biodiversity across the Eastern Mediterranean. It thus seems that, as with many large, slow-growing trees, oaks, cedars, junipers, and firs [11,13,48,51] may lack the ability to disperse efficiently into altered ranges fast enough to keep pace with the rapid change predicted. These findings and our insights about *Quercus alnifolia* reinforce the notion that even diverse Mediterranean plant taxa with extensive current geographic presence are under intense, climate-linked pressures in this climate change hotspot.

Furthermore, our preliminary evaluation of *Quercus alnifolia*’s risk of extinction, applying IUCN Categories A (decline in population size) and B (reduction in geographic range), categorises the species as Critically Endangered across all future periods and scenarios analysed (Appendix A). This classification is due to the expected reduction in range and heightened fragmentation. Should only IUCN Criterion B be considered, *Quercus alnifolia*’s risk status is expected to shift to Vulnerable, particularly as we approach the latter part of the 21st century.

### 3.3. Sensitivity, Exposure, and Vulnerability to Climate and Land-Use Change

Our sensitivity, exposure, and vulnerability analysis (Appendix A; Figure 10) provides important insights into how *Quercus alnifolia* might cope with expected environmental shifts associated with climate and land-use change. *Quercus alnifolia* displays lower vulnerability values compared to range-restricted *Magnolia* trees in Asia and the Americas [137], but higher than *Juniperus drupacea*, another Mediterranean range-restricted tree [11]. The moderate sensitivity classification is revealing when juxtaposed against the heightened exposure and vulnerability we find under more pessimistic emission scenarios. This suggests a complex interplay: some degree of tolerance for initial ecological disturbance may coexist with severe repercussions from more abrupt or pronounced changes. These findings underscore the urgency of limiting the further intensification of disruptive forces within the species’ range. *Quercus alnifolia* requires a swift and integrated overhaul of management strategies, policy frameworks, and conservation initiatives. Heightened vulnerability within Troodos National Park (Figure 10), the core population area, demands targeted, proactive conservation measures as environmental changes may outpace *Quercus alnifolia*’s inherent adaptability. Such proactive strategies may include assisted range shifts—even on limited scales—by facilitating dispersal for relic or rear-edge populations while monitoring threats in remaining core habitats.

Our findings highlight the dire threat facing even resilient species like *Quercus alnifolia*. If a long-lived, drought-tolerant tree struggles under climate and land-use changes, the outlook is far more severe for many Cypriot endemic plants, especially those with small, genetically depauperate populations within fragmented habitats, making them highly susceptible to extinction [138]. A critical need exists to assess these endemic plants’ climate and land-use change vulnerability. Similar analyses should be prioritised, supported by increased Cypriot collaboration with other EU and Mediterranean nations. This aligns with the pressing need to expand the extremely limited research on climate change impacts within Cyprus’ unique flora.

### 3.4. Conservation Implications and Measures

*Quercus alnifolia* colonises remote and mainly steep slopes, where other woody plant taxa cannot, and its communities prevent erosion and stabilise soil, making these habitats more “life-friendly” [58,59,66]. Furthermore, *Quercus alnifolia* forests contribute to the region’s biodiversity, supporting a variety of flora and fauna essential for ecological balance and tourism, which is a significant source of income for many communities. Tourists are drawn to the Troodos mountain range for its natural beauty, hiking trails, and the unique biodiversity it supports, including the picturesque *Quercus alnifolia* forests. A decline in this species would likely lead to a decrease in tourism appeal, adversely affecting local businesses and livelihoods reliant on this sector. Therefore, preserving *Quercus alnifolia* is a matter of ecological importance and economic sustainability for communities that depend on the ecosystem services it provides [29].

The potential loss of *Quercus alnifolia* and the subsequent impact on ecosystem stability in the Troodos region threaten crucial natural services underpinning human life across Cyprus, but even the decline of *Quercus alnifolia* would have profound socioeconomic impacts on local communities. As a primary component of erosion prevention and a key regulator of nutrient flow, forest health, and overall vegetation structure [61,62], *Quercus alnifolia* supports the water cycle and soil quality, upon which agricultural productivity, local industries, and water quality in adjacent areas depend. Its extensive root systems, by preventing soil erosion, maintain fertile lands for farming and safeguard against the damaging effects of landslides and floods, which can have devastating consequences for local economies and infrastructure. Moreover, with forest vegetation diminished and increasing fire occurrences, natural hydrological mechanisms that govern both slow water release and prevent dangerous runoff could be disrupted [139,140,141]. These alterations impact Cypriot communities’ drinking water access, flood risk management, and irrigation resources. Thus, *Quercus alnifolia*’s future holds direct significance for the health of broader, diverse ecosystems and the well-being of populations that rely on their environmental services.

Studies investigating the impacts of losing foundational tree species in other Mediterranean hotspots (whether from drought, fires, or pests) indicate adverse effects on overall species richness, trophic cascade disruptions, and degradation of habitats relied upon by a diverse range of flora and fauna. Our findings resonate strongly with broader, dire assessments of biodiversity threats faced by vulnerable Mediterranean endemic tree species [11,46,48,51]. Cyprus’ island environment amplifies those risks and underlines the urgent need for holistic responses—prioritising biodiversity conservation through integrated habitat protection alongside mitigation strategies that consider future ecological and human needs in tandem.

The case of *Quercus alnifolia* in Cyprus offers both a warning, prompting immediate, dedicated intervention, and an opportunity to develop model approaches for managing at-risk endemic tree species foundational to healthy Mediterranean ecosystems in this increasingly precarious climatic era.

Prioritising monitoring within established reserves is paramount to ensuring long-term protection and enhancing understanding. Assessments of population health, soil dynamics, seed dispersal, and the potential impacts of future erosion shifts provide a solid foundation for conservation. Since low genetic diversity has already been identified for *Quercus alnifolia*, research could focus on its implications for long-term climate adaptiveness and explore whether management intervention is required to promote resilient populations. Strategic interventions, like assisted migration and utilising climate-adapted varieties, may be needed to maintain resilience and minimise the risk of local extinction for *Quercus alnifolia* within the Troodos mountain range. Additionally, monitoring responses to changing fire regimes (both regarding frequency and intensity) within reserves is crucial, as fires influence regeneration and may pose an increasing threat given projected climate changes. While some degree of community adaptability within ecosystems is likely, the combination of range constraints and predicted increases in forest fires creates substantial risks for long-term biodiversity resilience. Given the high proportion of *Quercus alnifolia* within protected areas, policies that actively support these reserves deserve a high priority. Enforcing sustainable land management practices that avoid degradation in adjacent areas can help shield populations from indirect anthropogenic pressures and reduce human–wildlife conflict associated with range retraction.

The potential implications of the projected shifts in *Quercus alnifolia*’s distribution have profound ramifications for Cyprus’s broader ecological systems. As a critical member of the Troodos serpentine flora and an essential component of ultramafic maquis vegetation, this keystone species acts as a cornerstone, underpinning various aspects of biodiversity and ecosystem function. Climate change mitigation initiatives at a national level also offer cascading benefits through reduced pressures on vulnerable protected zones and could include incentives for maintaining or augmenting ecological functions that contribute to this endemic tree’s wellbeing.

Given the critical role of *Quercus alnifolia* in maintaining ecosystem services and its contribution to local economies, a holistic, strategic, and integrated approach to its conservation is imperative and should integrate sustainable land use, erosion control, national policy alignment, and citizen science. Prioritizing soil health is critical; reforestation with native species and agroforestry practices will protect existing habitat and encourage the spread of *Quercus alnifolia*. Targeted erosion control measures, such as gabions and check dams, must be deployed in vulnerable areas, particularly those predicted to be suitable for the species in the future.

Recognising the Plant Micro-Reserve (PMR) as a foundational element in *Quercus alnifolia* conservation, we advocate for a broader strategy to address the challenges posed by climate change, including range contraction, habitat vulnerability, and reduced genetic diversity.

To aid *Quercus alnifolia*’s resilience, we concur with the PMR’s [74] suggestion of a systematic monitoring programme that encompasses the entire range of the species to identify vulnerable areas and track population dynamics. This programme should include periodic assessments of species distribution, population density, and ecosystem health, adhering to established methodologies such as the Braun-Blanquet approach and IUCN guidelines. Such comprehensive monitoring will enable targeted conservation actions and inform policy decisions. Research focusing on genetic diversity and adaptive capacity is crucial for developing effective conservation strategies. Investigating seed banking viability and exploring assisted gene flow to enhance genetic diversity is essential to ensuring the species’ long-term survival [74]. National policies must be aligned with conservation objectives, promoting sustainable land use and incentivizing practices that support ecosystem services provided by *Quercus alnifolia*. Integrating conservation goals into the EU Biodiversity Strategy and pursuing financial support for targeted actions are critical for enabling effective conservation measures. Further, enhancing ecological connectivity by creating wildlife corridors and supporting seed dispersal research is vital for maintaining genetic flow and population viability [74]. Engaging the public through citizen science initiatives, such as the involvement of volunteers in monitoring, will raise awareness and contribute valuable data to conservation efforts [74].

Situated within the Mt. Troodos UNESCO Geopark, the botanical garden is vital to *Quercus alnifolia* conservation. Its unique position facilitates a dynamic interplay between research, education, and community engagement. Robust citizen science programmes focused on habitat mapping, tree health, seed dispersal, and studying climate impacts on *Quercus alnifolia* offer a two-pronged benefit: they engage the public and generate crucial data to inform conservation strategies. The UNESCO designation strengthens these efforts, providing channels for international collaboration, funding, and participation in global biodiversity protection initiatives. The botanical garden on Mt. Troodos or the Nature Conservation Unit situated in Nicosia could partner with local educators and professionals to develop targeted educational programs to cultivate future conservationists. These initiatives could embed conservation principles within the curriculum, fostering a sense of environmental stewardship from a young age. By championing sustainable practices and actively advocating for conservation-minded policies, the Mt. Troodos botanical garden and the Natural Conservation Unit have the potential to not only safeguard *Quercus alnifolia* but also inspire a model of collaborative conservation with broader impacts, building a committed alliance for environmental protection.

To address the broader implications of climate change, Cyprus should partner with international conservation organizations like the International Union for Conservation of Nature (IUCN). These partnerships could facilitate knowledge exchange, provide access to funding for conservation projects, and support the implementation of best practices in habitat management and species conservation. Integrating climate change adaptation measures into national biodiversity strategies is crucial on the policy front. Harmonising conservation actions with national strategies, like the Cyprus National Biodiversity Strategy and Action Plan, ensures that efforts to protect *Quercus alnifolia* are coherent with wider sustainability and climate adaptation objectives. This promotes efficient resource utilisation and reinforces policy support. Additionally, leveraging funding opportunities from EU programmes, such as LIFE or Horizon Europe, can provide vital financial support for these conservation measures, increasing their scale and effectiveness. Policies should encourage sustainable land use practices that reduce habitat fragmentation and promote ecological connectivity, enabling species migration and genetic flow.

Moreover, Cyprus could lead regional initiatives under the Barcelona Convention to develop Mediterranean-wide strategies addressing the impacts of climate change on endemic species. Such efforts would benefit *Quercus alnifolia* and bolster the resilience of Mediterranean ecosystems, ensuring their preservation for future generations.

## 4. Materials and Methods

### 4.1. Species Occurrence Data

The delineation of the potential distribution area for *Quercus alnifolia* was aligned with its Extent of Occurrence (Figure 11), conforming to the criteria established by the IUCN [67,142], in accordance with the IUCN’s established protocols [143]. The collection of occurrence data for *Quercus alnifolia* was systematically compiled from diverse sources, including:The Global Biodiversity Information Facility database documenting 92 occurrences [144];Comprehensive studies by Sotiriou and Gerasimidis (2009), Sotiriou (2010) (Ph.D. thesis), and Milios et al. (2021), together accounting for 139 occurrences [58,59,60];Research conducted by Constantinou & Panitsa (2023), detailing 36 occurrences [57];Data extracted from the EUForest database, indicating 8 occurrences [145];The WOODIV database, which provided occurrence data in 10 × 10 km grids, albeit with significantly less precision [146].

Aware of the limitations imposed by discarding data marked with positional uncertainty—namely a reduced sample size and compromised model precision, which could hinder accurate species distribution assessment—we adopted the framework recommended by [147] and developed by [148], particularly the Nearest Environmental Point method as implemented in the *enmSdmX* 1.1.2 R package [148], to counteract these limitations. This approach enabled the inclusion of an additional 23 occurrence points from the WOODIV database into our analysis, culminating in a comprehensive dataset encompassing 298 occurrences.

We subsequently refined our dataset by eliminating records dated before 1990 to ensure consistency with the temporal scope of our environmental data (see Section 4.2 Environmental Data). Entries exhibiting coordinate uncertainty exceeding 1000 m were also excluded. The data cleansing process was performed using the ’clean_coordinates’ function from the *CoordinateCleaner* 2.0.18 R package [149]. Further, we eliminated duplicate occurrences using the ’elimCellDups’ function from the *enmSdm* 0.5.3.3 R package [150]. The final step in data curation involved spatial thinning (at the same spatial resolution as the environmental data, i.e., 1000 m; see Section 4.2 Environmental Data) of the remaining data, implemented via the ‘thin’ function from the *spThin* 0.1.0 [151] R package, following the methodologies outlined in [151,152]. Thus, our final dataset consisted of 78 records for *Quercus alnifolia*.

### 4.2. Environmental Data

We compiled a comprehensive climate dataset from 1981 to 2009, incorporating the 19 standard bioclimatic variables identified by WorldClim v.2 [87,88,89]. We enriched this dataset by adding 16 environmental variables as specified in [86] (ENVIREM dataset) to reflect the temporal breadth of the collected occurrence data. We set the spatial resolution of the dataset at 1000 m and integrated altitude data using the ‘elevation_30s’ function from the *geodata* 0.5–8 [153] R package. We employed the ClimateEU v4.63 [154], *dismo* 1.1.4 [155], and *envirem* 2.2 [86] R packages for environmental data creation, strictly adhering to the methodologies and protocols documented in [154,156,157]. We sourced soil metrics from SoilGrids v.2 [84] and dynamic land use/land cover (LULC) data from Chen et al. [90] at the resolution of other environmental metrics. We calculated essential topographical metrics such as aspect, heat load index, slope, topographic position index, and terrain ruggedness index using altitude data and functions from the *raster* 2.6.7 [158], *terra* 1.7.46 R package [159], and *spatialEco* 1.2-0 R packages [160]. We also obtained rainfall soil erosivity data developed by Panagos et al. [91] and available from the European Soil Data Centre [161,162], ensuring alignment with the spatial resolution of other environmental variables. We also integrated soil hydraulic characteristics, including field capacity, moisture retention curve, saturated water content, wilting point, saturated hydraulic connectivity, and hydraulic conductivity curve [92], with the environmental dataset, maintaining spatial resolution consistency.

For future climate conditions, we delineated projections for two periods: the 2050s (2041–2070) and the 2080s (2071–2100), using projections from [154]. These projections utilized data from two global circulation models (CCSM4 and HadGEM2), two IPCC representative concentration pathways (RCP4.5 and RCP8.5), and future LULC projections from Chen et al. [90] under SSP1-RCP26, SSP3-RCP70, and SSP5-RCP85 scenarios [163].

We categorised 58 environmental variables, treating topographical, soil (sourced from SoilGrids v.2), and hydrological variables as invariant over time while considering bioclimatic, LULC, and soil erosivity variables as dynamic. We rigorously selected a subset of 20 environmental variables to minimize collinearity risks. We systematically validated this selection using Spearman rank correlation (<0.7) and variance inflation factors (<5) [164], employing the *collinear* 1.1.1 R package [165] for precise collinearity diagnostics.

### 4.3. Land Use and Land Cover Changes

The dynamics of land use and land cover and their alterations are continually monitored due to the recent significant increase in land use and cover changes [166]. We employed the *OpenLand* 1.0.2 package in R for this analysis; it provides a robust, integrated approach to exploring LUC alterations [166]. We examined and visualised land use and land cover alteration progression within our study region using this package. Additionally, we used the package to perform an intensity analysis on available LULC data. This analysis sought to determine the rate of land cover changes and to understand their transition dynamics.

### 4.4. Species Distribution Models

The occurrence-to-predictor ratio for *Quercus alnifolia* was below 10:1. We thus aligned with the guidelines in [167,168] to precisely model the realised climatic niche of *Quercus alnifolia*. We employed the Random Forest algorithm and the *ecospat* 3.1 [169] R package, as detailed in [170,171].

For generating pseudo-absences, we used the ’sample_pseudoabs’ function from the *flexsdm* 1.3.3 R package [172]. This process entailed creating pseudo-absences within a geographical buffer, environmentally constraining them, and distributing them in environmental space through k-means clustering [172,173].

We carried out an optimized spatial cross-validation of occurrences and pseudo-absences prior to model fitting, following the recommendations in [174,175], utilising the ’part_sblock’ function from the *flexsdm* 1.3.3 R package [172]. Subsequently, we assessed the model’s performance against null models [176], employing several metrics [177,178,179,180,181] as recommended by [182] using functions from the *CalibratR* 0.1.2, *DescTools* 0.99.40, *ecospat* 3.2, *enmSdm* 0.5.3.2, *Metrics* 0.1.4, *MLmetrics* 1.1.1, and *modEvA*2.0 R packages [150,169,183,184,185,186].

To enhance the robustness and depth of our model evaluation, we incorporated the comprehensive pooling procedure as outlined by Collart et al. [187], ensuring a more nuanced and detailed analysis of our models’ performance. We selected well-calibrated models with a minimum TSS score of 0.4 to delineate potential habitats for *Quercus alnifolia* for each analysis period.

For generating binary maps for every GCM, RCP, SSP, and period combination, we employed the metric optimizing sensitivity and specificity [188].

To address environmental and geographical extrapolation, we used the ‘Shape’ metric as implemented in the *flexsdm* 1.3.3 R package [172]. Following recommendations in [189], we explored a variety of distance thresholds to truncate our model’s predictions. This approach serves as a precautionary measure to mitigate potential prediction issues [190]. We employed the *humboldt* R package [104] to assess and address potential niche truncation within our predictive models.

We evaluated potential range shifts of *Quercus alnifolia* in terms of directionality and extent using the *biomod2* 4.2.4 R package [191]. To simulate realistic dispersal, we allowed *Quercus alnifolia* to migrate from its initial range into environmentally suitable regions based on its natural dispersal ability. This dispersal scenario was implemented in *MigClim* [192,193], an R package enabling the simulation of species dispersal, colonization, growth, and local extinction. For this analysis, truncated binary distributions for current and anticipated future scenarios were assimilated into *MigClim*, focusing on the 2050s and 2070s. We employed a 30-year dispersal step to approximate the reproductive age of *Quercus alnifolia*, following [194]. We modelled species dispersal by employing dispersal kernels, which integrate mean dispersal distances and the potential for propagule generation post-cell colonisation. Our approach utilised a dispersal kernel characterised by a negative exponential function, aligning with methodologies prevalent in studies on plant species distribution amid environmental changes [192,194]. Given the limited availability of species-specific mean dispersal distances, we estimated the mean dispersal distance of *Quercus alnifolia* from its maximum dispersal distance using the following equation from [77,195]:Mean dispersal distance = 10 ^ (log10(maximum dispersal distance) − 0.795)/0.984)

We obtained the maximum dispersal distance of *Quercus alnifolia* from [80]. Following [194], we set the *MigClim* parameters ‘maturity age’ and ‘propagule production’ to 1, as the former approximates the typical maturation period of many trees, and the latter reflects the assumption that all mature trees generate propagules [196]. To enhance model reliability and reduce stochastic variability, we iterated this procedure 100 times, subsequently averaging the outcomes. Species dispersal simulations help counterbalance a limitation inherent in SDMs based on both dynamic and static environmental factors. When static variables play a dominant role, predictions may suggest that future habitat is suitable simply because conditions like the slope have not changed—even if climate has made that area no longer hospitable. By modelling realistic dispersal patterns, we acknowledge that static variables cannot be the sole determining factor for future range dynamics.

Finally, we calculated four fragmentation metrics: the number of patches, the patch cohesion index, the patch area (min, max, mean, and standard deviation), and the effective mesh size [197], for every GCM, RCP, SSP, and period combination for *Quercus alnifolia* using functions from the *landscapemetrics* 2.0.0 R package [198].

### 4.5. Sensitivity Analysis

To ensure model robustness and dissect the distinct roles of environmental variables and dispersal constraints, we performed a multi-level sensitivity analysis incorporating diverse parameter combinations within our Species Distribution Modeling framework. We systematically tested these combinations to explore their impact on model outcomes:
Environmental outcomes:
Climate-only: 19 bioclimatic WorldClim variables and 16 environmental variables from ENVIREM;Climate + Topography: Climate-only data supplemented by topographical variables;Climate + Topography + Soil + Hydrology: Addition of soil and hydrological data to a prior combination;Climate + Topography + Soil + Hydrology + LULC: The comprehensive suite with land-use and land-cover data.
Dispersal scenarios:
No Dispersal: Assumption of static range without species movement to suitable habitat;Unlimited Dispersal: A hypothetical scenario of unhindered range shifts towards ideal conditions;Specific Dispersal Constraints: Implemented only for the full model (incorporating all environmental variables) to evaluate dispersal limitations based on *Quercus alnifolia* specific characteristics.

Exploring these parameter permutations allows us to assess not only the influence of model simplification within environmental variable selection but also to isolate the impacts of contrasting dispersal assumptions on our projections for *Quercus alnifolia*.

### 4.6. Sensitivity, Exposure, and Vulnerability to Climate and Land-Use Change

Utilising the climate niche factor analysis delineated in [199], we evaluated species’ sensitivity, exposure, and vulnerability to climate and land-use changes, with a particular focus on temporally dynamic variables, using the *CENFA* 1.1.2 R package [200]. Species sensitivity is defined as the extent to which a species’ persistence is influenced by the climatic conditions of its existing habitat [201]. Typically, species more constrained by their present climate conditions tend to exhibit greater sensitivity to expected climate changes [199]. Conversely, exposure measures the extent of climate change a species is likely to experience within its habitat. This metric is derived by assessing the dissimilarity between current and projected climatic conditions within the species’ habitat, with higher dissimilarity indicating more substantial changes between present and future climates [199]. We computed species sensitivity and exposure by applying the ‘cnfa’ and ‘departure’ functions from the *CENFA* 1.1.2 R package [200]. Considering the extended lifespans of tree species like *Quercus alnifolia*, which limit swift adaptation to climatic variations [202], vulnerability to climate change was calculated by taking the geometric mean of sensitivity and exposure using the “vulnerability” function in the *CENFA* R package.

### 4.7. Future IUCN Extinction Risk Assessment

For each combination of GCM, RCP, SSP, and period, we allocated a preliminary IUCN threat category to *Quercus alnifolia* in line with IUCN Criteria A and B. This process involved the use of the *ConR* 1.1.1 R package [203] and the application of the R code available from [204], building upon a methodology previously implemented across a broader geographical and taxonomical scope in Greece by [9,11,53,127]. Subsequently, we examined discrepancies between its projected and current IUCN extinction risk status, drawing on the projections of our models and their binary transformations. To facilitate a transparent comparison, we also estimated the current IUCN extinction risk status of *Quercus alnifolia*. The analyses mentioned above form the basis of this species’ preliminary extinction risk assessment.

## 5. Conclusions

This study highlights the urgent necessity for a multi-pronged conservation approach centred on *Quercus alnifolia*. Integrating research tailored to this species with proactive policies focused on climate resilience and landscape management is essential. Our analysis, incorporating land-use dynamics, hydrological shifts, soil erosion, and dispersal constraints, indicates that existing protected areas will become increasingly vital amidst projected habitat contractions. Strengthening these strongholds must be a top priority for *Quercus alnifolia* conservation. The potential for conservation measures like assisted migration or exploring drought tolerance should be evaluated in a realistic landscape context. Alongside in-situ protection, sustainable land-use strategies in adjoining areas are critical to lessening human–wildlife conflicts and buffering populations from external stressors. Additionally, nationwide climate change mitigation efforts will have cascading benefits for vulnerable reserves and overall ecosystem health.

The projected shifts in *Quercus alnifolia* distribution raise serious concerns for Cyprus’ ecological systems. Its status as a keystone species within Troodos serpentine flora and a key component of ultramafic maquis habitat means its decline would resonate across diverse ecological functions, mirroring vulnerabilities in other Mediterranean biodiversity hotspots. Research demonstrates that tree loss triggered by environmental pressures leads to decreased overall species richness, cascading trophic disruptions, and widespread habitat degradation. While inherent adaptability exists within ecosystems, combining range restrictions with intensifying wildfire threats places long-term biodiversity resilience at significant risk, necessitating proactive countermeasures.

Our findings illuminate the potential impacts of climate and land-use changes on *Quercus alnifolia*, underscoring the value of incorporating a wide range of variables for accurate projections of evolving environmental risks. These results reveal the potential fragility of this keystone endemic tree in the Cypriot Troodos Mountains, compelling, targeted conservation action, and a comprehensive plan to protect its future.

Our study establishes a model for investigating other significant endemic species. Our comprehensive data on vulnerability factors and their impact on Cypriot ecological systems highlights the interconnected complexities of the Mediterranean landscape. Successful conservation endeavours must integrate insights from species-specific research into policies and initiatives designed to protect Cyprus’ unique botanical heritage, thus promoting resilient and robust ecosystems within this critical biodiversity hotspot.

## Figures and Tables

**Figure 1 plants-13-01109-f001:**
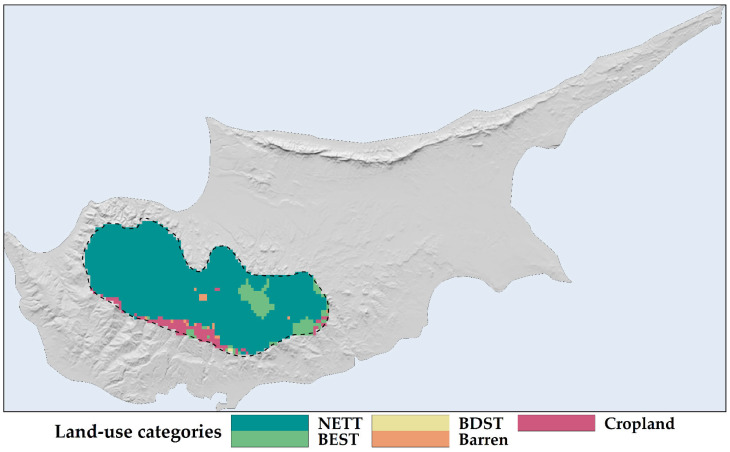
Baseline land-use types in the study area. Acronyms: BDST (broadleaf deciduous temperate shrubs), BEST (broadleaf evergreen temperate shrubs), NETT (needleleaf evergreen trees; conifers). The black dashed line delineates the IUCN-defined potential distribution area of *Quercus alnifolia*.

**Figure 2 plants-13-01109-f002:**
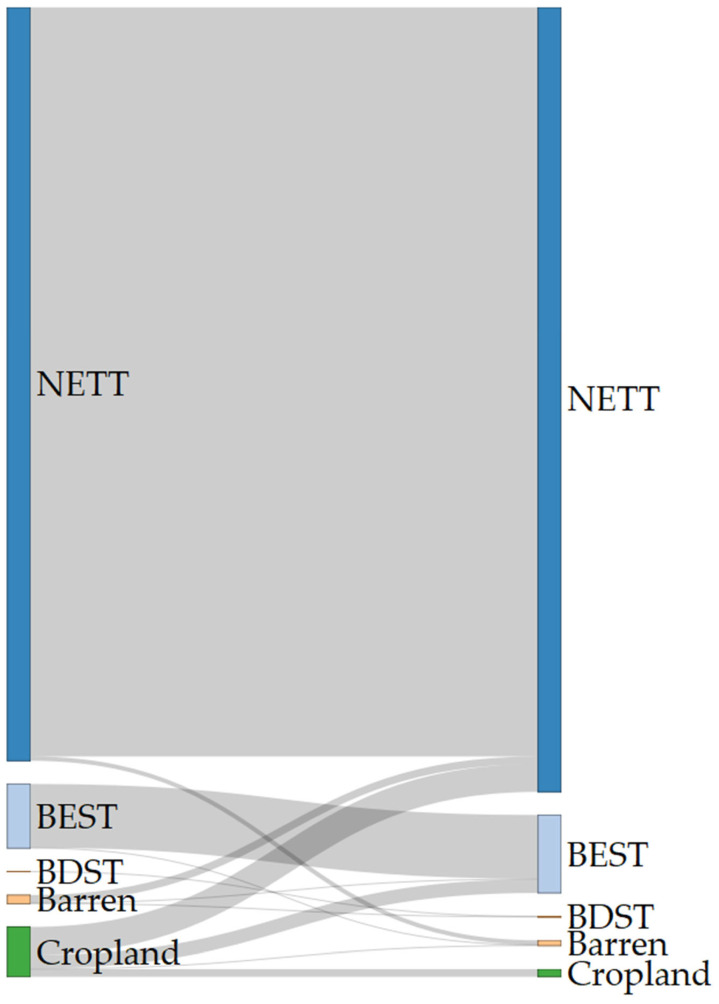
Projected land cover changes from 2015 to 2100 under Shared Socioeconomic Pathway 5 in the study area. Bars show the area occupied by each land cover type in 2015 (**left**) and 2100 (**right**). Flows illustrate the expected transition paths between these periods. Acronyms: BDST (broadleaf deciduous temperate shrubs), BEST (broadleaf evergreen temperate shrubs), NETT (needleleaf evergreen trees; conifers).

**Figure 3 plants-13-01109-f003:**
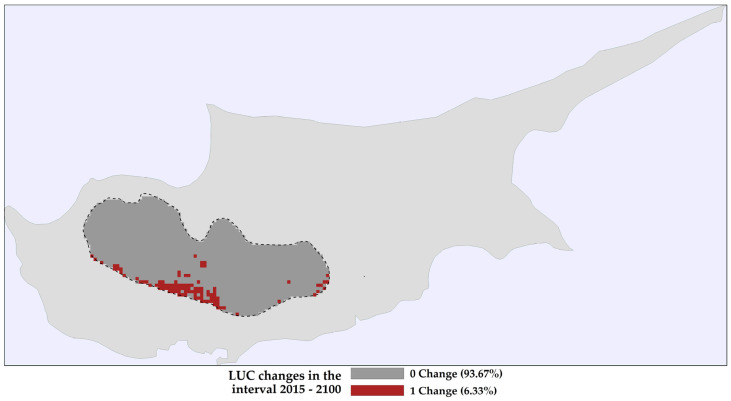
The anticipated number of land use and land cover changes (LUC) in the study area between 2015 and 2100 under the Shared Socioeconomic Pathway 5. The black dashed line consistently outlines the IUCN-defined potential distribution area of *Quercus alnifolia*.

**Figure 4 plants-13-01109-f004:**
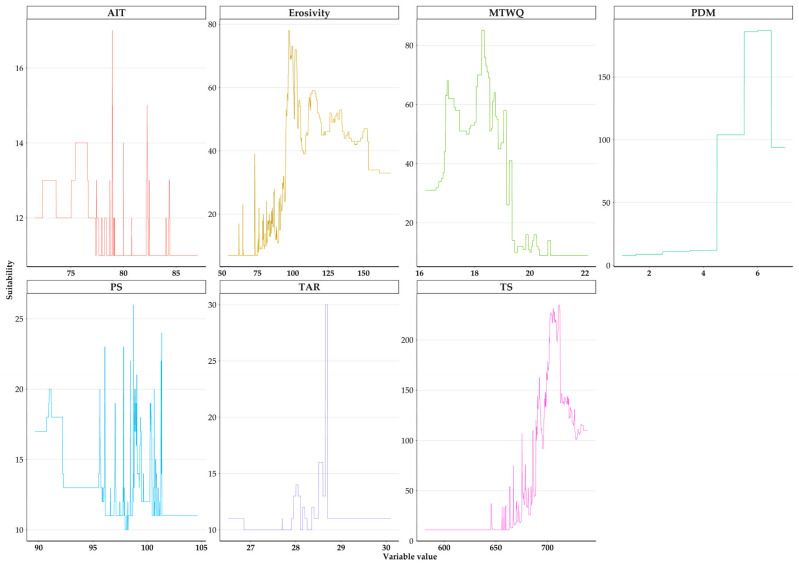
Response curves of temporally dynamic bioclimatic and soil variables. This figure presents the response curves for key bioclimatic and soil variables affecting *Quercus alnifolia*, as analysed in our study. Variables include AIT (Thornthwaite’s Aridity Index), MTWQ (Minimum Temperature of the Warmest Quarter), PDM (Precipitation of the Driest Month), PS (Precipitation Seasonality), TAR (Temperature Annual Range), and TS (Temperature Seasonality). Suitability scores for *Quercus alnifolia* are quantified on a scale from 0 to 1000, indicating the range of habitat suitability.

**Figure 5 plants-13-01109-f005:**
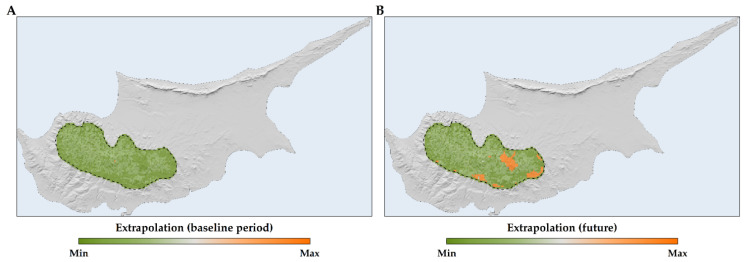
Patterns of extrapolation distance for projection data using the Shape metric in geographic space. Panel (**A**) represents the baseline period, while panel (**B**) depicts the future period (2070s under the HadGEM2 RCP 8.5 SSP5 scenario). In both panels, the potential distribution area of *Quercus alnifolia*, as defined by the IUCN, is delineated by a consistent black dashed line.

**Figure 6 plants-13-01109-f006:**
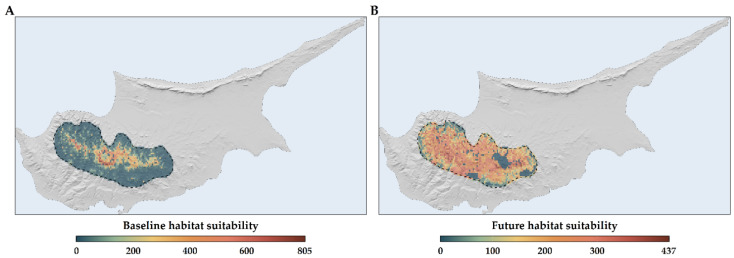
Habitat suitability for *Quercus alnifolia*: present and projected. Panel (**A**) displays the current habitat suitability, while panel (**B**) illustrates the projected suitability for the 2070s under the HadGEM2 RCP 8.5 SSP5 scenario. The maps depict continuous habitat suitability values, which were subsequently converted to binary maps (presence/absence) using a threshold value of 205, optimizing sensitivity and specificity. The potential distribution area of *Quercus alnifolia*, as outlined by the IUCN, is demarcated by a black dashed line.

**Figure 7 plants-13-01109-f007:**
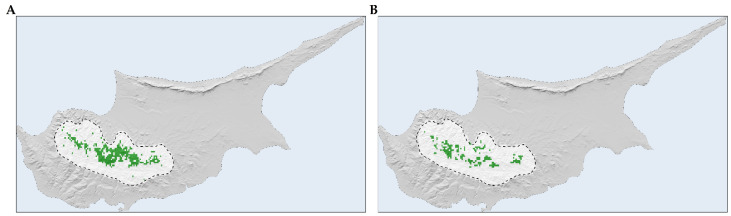
Binary maps illustrating habitat suitability for *Quercus alnifolia*: present and future projections. Panel (**A**) depicts the current habitat suitability, while panel (**B**) presents the projected suitability for the 2070s, derived from the HadGEM2 RCP 8.5 SSP5 scenario. These binary maps were generated using a threshold value of 205, optimizing sensitivity and specificity, with green cells denoting the presence of the species. In both panels, a black dashed line consistently outlines the IUCN-defined potential distribution area of *Quercus alnifolia*.

**Figure 8 plants-13-01109-f008:**
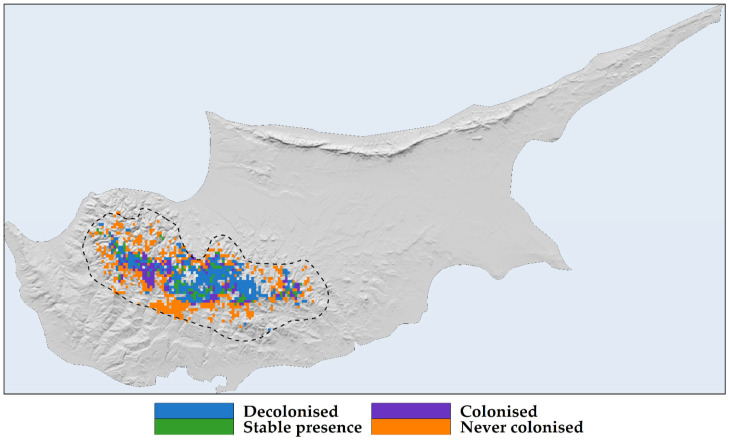
Range change map for *Quercus alnifolia* in the 2070s. This map visualizes the anticipated changes in the distribution of *Quercus alnifolia* under the HadGEM2 RCP 8.5 SSP5 scenario within the IUCN-defined potential distribution area, based on the *MigClim* dispersal constraints. Grid cells are colour-coded to represent changes in habitat suitability. Orange cells indicate areas that, while becoming climatically suitable, face natural colonisation barriers due to dispersal constraints; green cells denote areas where the species is currently found and is expected to persist in the future; grey cells represent areas where the species is neither currently present nor projected to be present in the future; blue cells designate areas where the species is currently present but is projected to be absent in the future; purple cells mark areas where the species is not currently present but is projected to be present in the future. The black dashed line consistently outlines the IUCN-defined potential distribution area of *Quercus alnifolia*.

**Figure 9 plants-13-01109-f009:**
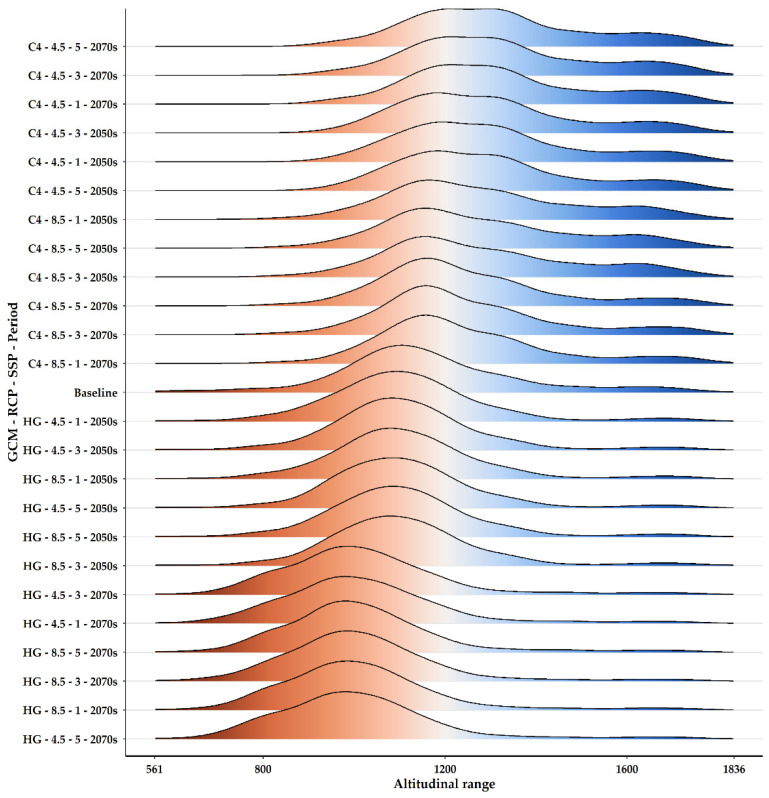
Altitudinal distribution of *Quercus alnifolia* across GCMs, RCPs, SSPs, and periods. This ridge plot illustrates the range of altitudes at which *Quercus alnifolia* occurs, as projected by various Global Circulation Models (GCMs), under different climate scenarios [Representative Concentration Pathways, RCPs (4.5 and 8.5), and Shared Socioeconomic Pathways, SSPs (1, 3, and 5)] across all periods analysed. Acronyms used: C4: CCSM4; HG: HadGEM2.

**Figure 10 plants-13-01109-f010:**
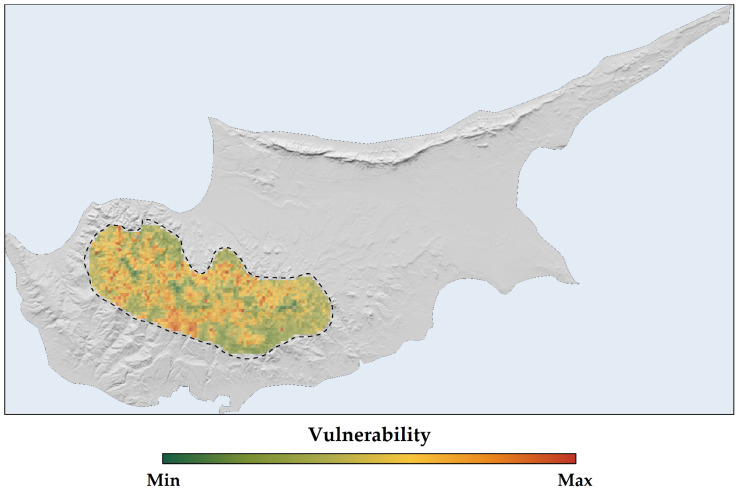
Spatial vulnerability of *Quercus alnifolia* in the 2070s under the HadGEM2 RCP 8.5 SSP5 scenario. The black dashed line consistently outlines the IUCN-defined potential distribution area of *Quercus alnifolia*.

**Figure 11 plants-13-01109-f011:**
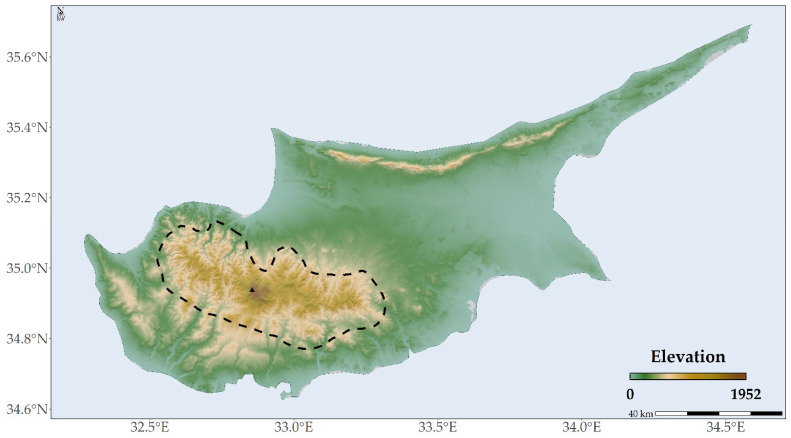
Topographical map of Cyprus. The potential distribution area of *Quercus alnifolia*, as defined by the IUCN, is demarcated by a black dashed line. A black triangle marks the highest peak of Mt. Troodos.

## Data Availability

Data are contained within the article and Appendix A.

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
