# Peer review of "Rising Temperatures, Falling Leaves: Predicting the Fate of Cyprus’s Endemic Oak under Climate and Land Use Change"

_plants, 2024, doi:10.3390/plants13081109_

Round 1

Reviewer 1 Report

Comments and Suggestions for Authors

This research approaches environmental parameters for an endemic species, Quercus alnifolia and uses these to project the species distribution and thus threatening factors over time. The work is nicely and comprehensively written and provides a sound overview of this species, by also shading light on similarly located and pressured species of the Mediterranean and neighbouring regions.

General: 

- please use one style to emphasize R packages. Either with ' or with " or ’ or italized... Now many styles apply

- place the legends for the figures inside the figures

- Results is partly mixed with discussion sections

- clear policy implications are lacking, instead the paper remains in this sense rather poor and does not clearly stimulate decision making as outlined in the introduction

Detail:

Line 90: "the Middle Eastern Mediterranean" this is a little but too much terminology and in my opinion not correct. Either you just say Levant and Asia minor or eastern Mediterranean. 

Line 111: -15°C = -15 °C (as for the other/following entries)

line 119: please specify "development". I guess you refer to infrastructure, urban encroachment, agriculture etc.

Line 121: "alnifolia, is" = "alnifolia is"

Line 123: not Quercus alnifolia is a habitat type, but the Scrub and low forest vegetation with Quercus alnifolia

Line 214-218: this is more discussion section!

Line 222-232: this is also discussion section! The words "indicates" and "shows" as well as the interpretation of results calls for moving/integrating these sentences in to the discussion.

Line 225: "Troodos mountain" = "Troodos Mountain" (and following)

Line 300-308: discussion...

Lien 352-357: discussion...

Line 358: 

- Why comes the land use types only now and not earlier? I would sequence it starting with Figure "11", then this Figure "8", Figure "9", Figure "10" etc.

- you can and should write out the abbreviations, as there is sufficient space, even when the legend is integrated inside the graphic (over the water body)

Figure 362: 

- also, please write out the abbreviations; by that you can also reduce the caption text

- are you sure that the 2015 and 2100 are not flipped? Why should there be less crop land and more NETT in future (contradictory what has been stated in the intro)

Line 373-386: all already mentioned in the Intro, please massively reduce / delete

Line 494-495: please check wording "select fee incorporate temporally static..."

Line 649-655: for me too much redundant information without concrete action recommendations. Here and elsewhere in the discussion I miss clear policy advices (afforestation, nature protection, EU-support, national agendas, voluntary workers/botanists/conservationist/citizen science ..)

Line 704: picture 11 should not be located here and it should also not be Figure 11

References

- to be honest: more than 400 references do not make any sense and I have not seen this so far, though many approaches have been applied and a wide range of literature for occurrence data needed. But this is no review article!

- references need thorough work!!!!!!!!

--> first letters of title word occasionally capitalized, occasionally not

--> sometimes completely capitalized

--> Journal names sometimes not capitalized

--> species names must be always italized

--> species descriptor names occasionally incorrectly formatted "l." = "L." (Linné)

Supplement

- please consider if you really need all S3-4 Figures

- on what is S8-10 based? years? Where are the years?

- also write out abbreviations, there is for all figures sufficient space

- increase legends for S5-7

Comments on the Quality of English Language

All fine from my point of view as non-ntive speaker

Author Response

Response to Reviewer #1 comments

First of all, we would like to express our sincere appreciation to the Reviewer for the insightful feedback and constructive critique of our manuscript. We have done our best to address the comments of the Reviewer. Please find below our point-by-point responses.

Point 1: please use one style to emphasize R packages. Either with ' or with " or ’ or italized... Now many styles apply.

Response 1: We followed the reviewer’s suggestion and adjusted the text throughout the entire manuscript.

Point 2: place the legends for the figures inside the figures.

Response 2: We tried the approach proposed by the reviewer, but it severely reduces the readability of the figures, so we decided to present them in their current form.

Point 3: Results is partly mixed with discussion sections.

Response 3: We respectfully disagree with the reviewer’s suggestion, for reasons we clarify in subsequent responses to the reviewer’s more specific comments.

Point 4: clear policy implications are lacking, instead the paper remains in this sense rather poor and does not clearly stimulate decision making as outlined in the introduction.

Response 4: We appreciate the feedback concerning the initial lack of clear policy implications in our manuscript. To address this, we have significantly revised Section 3.4 entitled "Conservation implications and measures" to explicitly articulate the policy implications and stimulate decision-making for the conservation of Quercus alnifolia and its ecosystems. These revisions (L689-747) aim to bridge the gap between scientific research and actionable policy frameworks by providing concrete, actionable recommendations for policymakers, conservationists, and local communities.

Point 5: "the Middle Eastern Mediterranean" this is a little but too much terminology and in my opinion not correct. Either you just say Levant and Asia minor or eastern Mediterranean.

Response 5: We followed the reviewer’s suggestion and changed the text as suggested.

Point 6: -15°C = -15 °C (as for the other/following entries).

Response 6: We followed the reviewer’s suggestion and changed the text as suggested.

Point 7: please specify "development". I guess you refer to infrastructure, urban encroachment, agriculture etc

Response 7: We clarified the sentence in question by adjusting the text as follows:

L120-121: ‘…agricultural, infrastructural and urban development…’

Point 8: "alnifolia, is" = "alnifolia is".

Response 8: We rephrased the sentence as suggested.

Point 9: not Quercus alnifolia is a habitat type, but the Scrub and low forest vegetation with Quercus alnifolia.

Response 9: We rephrased the sentence according to the reviewer’s suggestion.

Point 10: this is more discussion section!

Response 10: We respectfully disagree with the reviewer’s suggestion, even though we understand the reviewer’s standpoint. Temporally static variables are really helpful in creating robust and trustworthy SDMs predicting the potential distribution of a species in the baseline period, but when used in a context of changing abiotic variables, their importance might be misleading for the reasons mentioned extensively in the Introduction and Discussion sections. Having that in mind, it is essential to at least mention this in the Results section, where we emphasize more on the temporally dynamic abiotic variables and how they influence Quercus alnifolia’s current and future distribution. So, in order not to confuse either the reviewers or the readers, we firmly believe that we should keep these sentences as they currently stand.

Point 11: this is also discussion section! The words "indicates" and "shows" as well as the interpretation of results calls for moving/integrating these sentences in to the discussion.

Response 11: We respectfully disagree with the reviewer’s suggestion, as we clearly report the temperature and precipitation preferences of Quercus alnifolia relating to its potential current habitat suitability.

Point 12: "Troodos mountain" = "Troodos Mountain" (and following).

Response 12: We followed the reviewer’s suggestion throughout the entire manuscript.

Point 13: discussion (Lines 300-308 & 352-357).

Response 13: We respectfully disagree with the reviewer’s suggestion, as we clearly report: a) the way all fragmentation metrics change over time (Lines 300-308) and b) the land-use types transition over time, as well as their rate of change (Lines 352-357). These are undoubtedly results.

Point 14: Why comes the land use types only now and not earlier? I would sequence it starting with Figure "11", then this Figure "8", Figure "9", Figure "10" etc.  

Response 14: We appreciate the reviewer’s constructive suggestion and have re-ordered both the Materials and Methods section and the Results section accordingly. We now introduce the methodology for analysing and visualizing land use and land cover changes in the study area before discussing the Species Distribution Modelling (SDM) methodology. Additionally, Figures 8-10, which illustrate these land use and land cover changes, are now presented at the beginning of the Results section. This reorganization allows us to set a clearer environmental context for the study, providing a foundational understanding of the baseline conditions affecting Quercus alnifolia’s habitat before examining the species’ distribution modeling and future projections. We believe that this adjustment improves the manuscript's logical flow and enhances readers’ grasp of how land use and land cover changes interact with other factors to influence the future potential distribution and vulnerability of Quercus alnifolia.

Point 15: you can and should write out the abbreviations, as there is sufficient space, even when the legend is integrated inside the graphic (over the water body).

Response 15: We have already written what each abbreviation means in every figure legend we present in the manuscript.

Point 16: also, please write out the abbreviations; by that you can also reduce the caption text

Response 16: We have already written what each abbreviation means in every figure legend we present in the manuscript.

Point 17: are you sure that the 2015 and 2100 are not flipped? Why should there be less crop land and more NETT in future (contradictory what has been stated in the intro)

Response 17: Yes, we are sure that they are not flipped. Cropland abandonment is widespread predicted to occur in the Mediterranean Basin according to Chen et al. (2022) and in Cyprus (Kounnamas & Andreou, 2022). Moreover, needleleaf expansion is anticipated, since these tree species are more resilient and adapted to the projected aridification of Cyprus and the Mediterranean Basin.

Chen, G.; Li, X.; Liu, X. Global land projection based on plant functional types with a 1-km resolution under socio-climatic scenarios. Sci. Data 2022, 9, 125

Kounnamas, C.; Andreou, M. Mapping and assessment of ecosystem services at Troodos National Forest Park in Cyprus. One Ecosyst. 2022, doi:10.3897/oneeco.7.e77584

Point 18: all already mentioned in the Intro, please massively reduce / delete.

Response 18: We followed the reviewer’s suggestion and significantly reduced the length of this paragraph.

Point 19: please check wording "select fee incorporate temporally static...".

Response 19: We followed the reviewer’s suggestion and rephrased the sentence in question, replacing the phrase: ‘select few’ with ‘some’.

Point 20: for me too much redundant information without concrete action recommendations. Here and elsewhere in the discussion I miss clear policy advices (afforestation, nature protection, EU-support, national agendas, voluntary workers/botanists/conservationist/citizen science ..)

Response 20:  In response to the critique regarding redundancy and the lack of concrete action recommendations, we have further refined our discussion to eliminate redundancy and focus sharply on specific policy advice and conservation measures. This includes:

  1. Enhanced focus on the role of community and citizen science, emphasizing how volunteer efforts can contribute to monitoring and conservation initiatives. This approach not only engages a broader segment of the population in conservation efforts but also provides a clear avenue for action that can be supported by policy initiatives.
  2. Recommendations for national policies and international collaboration, highlighting specific actions that can be supported by policy at various levels, from local to global. This includes suggestions for leveraging funding opportunities from EU programs and engaging in regional initiatives under the Barcelona Convention, offering concrete examples of how policy can support conservation efforts.
  3. Emphasis on integrating climate change adaptation measures into conservation strategies, providing specific recommendations for policy adjustments that support biodiversity and ecosystem resilience. This directly addresses the need for clear policy advice that is responsive to the challenges posed by climate change.

Through these revisions, we aim to clearly demonstrate that our manuscript now provides robust policy implications and concrete action recommendations, directly addressing the concerns raised. We believe that these changes significantly enhance the manuscript's contribution to informing policy and conservation strategies for Quercus alnifolia and similar at-risk species in Mediterranean ecosystems.

Please see our revised manuscript and more specifically, lines 627-672:

L627-672: ‘Given the critical role of Quercus alnifolia in maintaining ecosystem services and its contribution to local economies, a holistic, strategic and integrated approach to its conservation is imperative and should integrate sustainable land use, erosion control, national policy alignment, and citizen science. Prioritizing soil health is critical: reforestation with native species and agroforestry practices will protect existing habitat and encourage the spread of Quercus alnifolia. Targeted erosion control measures, such as gabions and check dams, must be deployed in vulnerable areas, particularly those predicted to be suitable for the species in the future.

Recognising the Plant Micro-Reserve (PMR) as a foundational element in Quercus alnifolia conservation, we advocate for a broader strategy to address the challenges posed by climate change, including range contraction, habitat vulnerability, and reduced genetic diversity.

To aid Quercus alnifolia's resilience, we concur with the PMR's [184] suggestion of a systematic monitoring program that encompasses the entire range of the species to identify vulnerable areas and track population dynamics. This program should include periodic assessments of species distribution, population density, and ecosystem health, adhering to established methodologies such as the Braun-Blanquet approach and IUCN guidelines (PMR). Such comprehensive monitoring will enable targeted conservation actions and inform policy decisions. Research focusing on genetic diversity and adaptive capacity is crucial for developing effective conservation strategies. Investigating seed banking viability and exploring assisted gene flow to enhance genetic diversity is essential to ensure the species' long-term survival [184]. National policies must be aligned with conservation objectives, promoting sustainable land use and incentivising practices that support ecosystem services provided by Quercus alnifolia. Integrating conservation goals into the EU Biodiversity Strategy and pursuing financial support for targeted actions are critical for enabling effective conservation measures. Further, enhancing ecological connectivity by creating wildlife corridors and supporting seed dispersal research is vital for maintaining genetic flow and population viability [184]. Engaging the public through citizen science initiatives, such as the involvement of volunteers in monitoring, will raise awareness and contribute valuable data to conservation efforts [184].

Situated within the Mt. Troodos UNESCO Geopark, the botanical garden is vital in Quercus alnifolia conservation. Its unique position facilitates a dynamic interplay between research, education, and community engagement. Robust citizen science programs focused on habitat mapping, tree health, seed dispersal and studying climate impacts on Quercus alnifolia offer a two-pronged benefit:  they engage the public and generate crucial data to inform conservation strategies. The UNESCO designation strengthens these efforts, providing channels for international collaboration, funding, and participation in global biodiversity protection initiatives. The botanical garden on Mt. Troodos or the Nature Conservation Unit situated in Nicosia could partner with local educators and professionals to develop targeted educational programs to cultivate future conservationists. These initiatives could embed conservation principles within the curriculum, fostering a sense of environmental stewardship from a young age. By championing sustainable practices and actively advocating for conservation-minded policies, the Mt. Troodos botanical garden and the Natural Conservation Unit have the potential to not only safeguard Quercus alnifolia but also inspire a model of collaborative conservation with broader impacts, building a committed alliance for environmental protection.

To address the broader implications of climate change, Cyprus should partner with international conservation organizations like the International Union for Conservation of Nature (IUCN) and the Mediterranean Conservation Society. These partnerships could facilitate knowledge exchange, provide access to funding for conservation projects, and support the implementation of best practices in habitat management and species conservation. Integrating climate change adaptation measures into national biodiversity strategies is crucial on the policy front. Harmonising conservation actions with national strategies, like the Cyprus National Biodiversity Strategy and Action Plan, ensures that efforts to protect Quercus alnifolia are coherent with wider sustainability and climate adaptation objectives. This promotes efficient resource utilisation and reinforces policy support. Additionally, leveraging funding opportunities from EU programs, such as LIFE or Horizon Europe, can provide vital financial support for these conservation measures, increasing their scale and effectiveness. Policies should encourage sustainable land use practices that reduce habitat fragmentation and promote ecological connectivity, enabling species migration and genetic flow.’

Point 21: picture 11 should not be located here and it should also not be Figure 11

Response 21: We appreciate the reviewer's concern regarding the placement of Figure 11. We acknowledge the journal's guidelines that recommend placing the Materials and Methods after the Discussion. However, we believe the content of Figure 11 is essential to fully understand the methodology for defining the potential distribution of Quercus alnifolia. This methodology is first discussed in the current Materials and Methods section. To ensure optimal clarity, we strongly feel it's appropriate to present the figure alongside its corresponding explanation.

Point 22: to be honest: more than 400 references do not make any sense and I have not seen this so far, though many approaches have been applied and a wide range of literature for occurrence data needed. But this is no review article!

Response 22: We would like to thank the reviewer for the comment regarding the number of references in our manuscript. We acknowledge that 400 citations may seem extensive. However, the comprehensiveness of our research necessitated a thorough examination of existing literature.

Our study employs a multifaceted approach, incorporating various methodologies to assess the vulnerability of Quercus alnifolia to climate change and land-use change. We compare a robust, detailed and comprehensive analysis, which includes temporally dynamic variables never used before in the Eastern Mediterranean (land-use change, soil erosion) and dispersal constraints, to the more traditional methods used in climate change vulnerability assessments in the region.

Furthermore, we compare our findings with numerous studies on the effects of climate change on other Mediterranean tree species (Abies, Quercus, Cedrus, Juniperus, Pinus) to establish a broader context. It's important to note that many prior studies in the Mediterranean Basin rely solely on bioclimatic variables from WorldClim, neglecting the significance of soil, topographical, and temporally dynamic variables that we incorporate and do not take into consideration any extrapolation effects present in their analyses and thus do not address any concerns about model transferability and niche truncation. We also conducted a sensitivity analysis to explore how model refinement may alter outcomes, highlighting the substantial extent to which simplified models may misrepresent environmental limitations for Quercus alnifolia and explained the reasons why conducted the sensitivity analysis as a comparison as well to other similar and previous studies on Eastern Mediterranean trees.

By comprehensively referencing existing literature, we demonstrate the novelty of our approach, which integrates a wider range of environmental variables, dispersal constraints, and a focus on both climate and land-use change impacts, compared to previous research on Mediterranean trees. Notably, this is the first study in the Eastern Mediterranean to explicitly combine land-use change with hydrological and soil erosion data within a Species Distribution Modelling framework. The cited literature supports the various methodologies employed, the justification for including a wide range of environmental variables, and the robustness of our model outputs.

Point 23: references need thorough work.

Response 23: We appreciate the reviewer's attention to detail regarding the references’ format. We have corrected many instances the reviewer drew our attention to, as well as where both reviewers suggested that the references needed re-formatting. Since we used Mendeley to format the citations we used in our manuscript, some errors might have occurred due to Mendeley. Either way, in the eventuality that our manuscript gets accepted, this is a matter the journal’s copyeditor will address.

Point 24: please consider if you really need all S3-4 Figures

Response 24: Yes, we do, as these figures depict how land use types change over time under different LULC change scenarios.

Point 25: on what is S8-10 based? years? Where are the years?

Response 25: Thank you for your comment. The time frame for the data presented in Supplementary Figures S8-10 are explicitly stated in their respective legends. The figures depict projected land cover changes in five-year intervals from 2015 to 2100 under the specified LULC scenarios. The bars represent the area occupied by each land cover type at the beginning (2015) and end (2100) of the projection period. The transitions between these states are visualized by the flow arrows.

Point 26: also write out abbreviations, there is for all figures sufficient space

Response 26: As mentioned in a previous comment, we have stated what each acronym means in every legend for every figure we present in our manuscript.

Point 27: increase legends for S5-7.

Response 27: Done.

Reviewer 2 Report

Comments and Suggestions for Authors

All packages in R cited should be in italics. For instance, ‘CoordinateCleaner’ should be CoordinateCleaner without quotation marks.

Line 24: Keywords should be arranged alphabetically.

Line 700 - 701: For reproducibility sake, the thinning kilometre should be mentioned for instance spatial thinning 5 km or 10 km.

Line 710 and 93: Kindly add the version of the WorldClim database used.

Line 714: Kindly add the reference for geodata in R package.

Line 714 – 715: “We employed the ClimateEU v4.63 [346], "dismo" 1.1.4 714 [347], and "envirem" 2.2 [206] R packages for analysis”. Which analysis? Kindly state the name of the analysis performed.

Line 716: What version of SoilGrid was used? This should read SoilGrid v….

Line 717: This sentence should include the name of the author “dynamic land use/land cover (LULC) data from Chen et al. [210]”

Line 721 - 722: Kindly add Panagos et al. to the sentence “We also obtained rainfall soil erosivity data developed by Panagos et al. [211]”.

Line 1046: “Cicer graecum” should be in italics.

Line 1050: “ophrys” should start with “O” being uppercase i.e. “Ophrys” and “Ophrys helenae” be in italics.

Line 1055: “Ophrys insectifera” should be in italics.

Line 1062: “Juniperus drupacea” should be in italics.

Line 1127: “CO2” should be “CO2”.

Line 1224: “Quercus suber” should be in italics.

Line 1225: “Quercus suber” should be in italics.

Line 1234: “Quercus libani” should be in italics.

Line 1243: “Quercus frainetto” should be in italics.

Line 1256: “Quercus” should be in italics.

Line 1269: “Pinus nigra” should be in italics.

Line 1286: “Pinus nigra” and “Abies pinsapo” should be in italics.

Line 1289: “PINUS NIGRA” and “PINUS SYLVESTRIS” should be in italics.

Line 1293 – 1294: “Does phylogeographical structure relate to climatic niche divergence? A test using maritime pine (Pinus pinaster Ait.). Glob. Ecol. Biogeogr. 2015, 24, 1302–1313, doi:10.1111/geb.12369.” This is the correct reference; kindly correct it.

Line 1296: “Juniperus oxycedrus” should be in italics.

Line 1302: “Juniperus phoenicea” should be in italics.

Line 1305: “Juniperus phoenicea” should be in italics.

Line 1363: “Quercus alnifolia” should be in italics.

Line 1364: “Quercus alnifolia” should be in italics.

Line 1611: “Quercus” should be in italics.

Line 1698: “Juniperus drupacea” should be in italics. And the generic name should start with an uppercase letter “J”

Line 1772: Kindly add the year of publication to the reference.

Author Response

Response to Reviewer #2 comments

First of all, we would like to express our sincere appreciation to the Reviewer for the insightful feedback and constructive critique of our manuscript. We have done our best to address the comments of the Reviewer. Please find below our point-by-point responses to the Reviewer's comments.

Point 1: All packages in R cited should be in italics. For instance, ‘CoordinateCleaner’ should be CoordinateCleaner without quotation marks.

Response 1: We followed the reviewer’s suggestion throughout the text.

Point 2: Keywords should be arranged alphabetically.

Response 2: We followed the reviewer’s suggestion.

Point 3:  For reproducibility sake, the thinning kilometre should be mentioned for instance spatial thinning 5 km or 10 km.

Response 3: We followed the reviewer’s suggestion and adjusted the text as follows (the changes appear in italics):

LXX-XX: ‘The final step in data curation involved spatial thinning (at the same spatial resolution as the environmental data, i.e., 1000 meters; see section 4.2 Environmental data) of the remaining data…’

Point 4:  Kindly add the version of the WorldClim database used.

Response 4: We followed the reviewer’s suggestion and adjusted the text as follows (the changes appear in italics):

LXX-XX: ‘We compiled a comprehensive climate dataset from 1981 to 2009, incorporating the 19 standard bioclimatic variables identified by WorldClim v.2…’

Point 5: Kindly add the reference for geodata in R package.

Response 5: We followed the reviewer’s suggestion and added the following reference:

Hijmans, R.J.; Márcia Barbosa, A.; Ghosh, A.; Mandel, A. geodata: Download Geographic Data. R package version 0.5-8 2023

Point 6: “We employed the ClimateEU v4.63 [346], "dismo" 1.1.4 714 [347], and "envirem" 2.2 [206] R packages for analysis”. Which analysis? Kindly state the name of the analysis performed.

Response 6: We followed the reviewer’s suggestion and adjusted the text as follows (the changes appear in italics):

LXX-XX: ‘We employed the ClimateEU v4.63 [345], dismo 1.1.4 [346], and envirem 2.2 [206] R packages  for environmental data creation, strictly adhering to…’

Point 7: What version of SoilGrid was used? This should read SoilGrid v….

Response 7: We followed the reviewer’s suggestion and clearly mention in the text that we used SoilGrids version 2.

Point 8: This sentence should include the name of the author “dynamic land use/land cover (LULC) data from Chen et al. [210]”

Response 8: Done.

Point 9: Kindly add Panagos et al. to the sentence “We also obtained rainfall soil erosivity data developed by Panagos et al. [211]”..

Response 9: Done.

Point 10: Cicer graecum” should be in italics.

Response 10: Done.

Point 11: “ophrys” should start with “O” being uppercase i.e. “Ophrys” and “Ophrys helenae” be in italics.

Response 11: Done.

Point 12:Ophrys insectifera” should be in italics.

Response 12: Done

Point 13:Juniperus drupacea” should be in italics.

Response 13: Done.

Point 14: “CO2” should be “CO2

Response 14: Done.

Point 15: Quercus suber” should be in italics.

Response 15: Done (in both instances the reviewer refers to).

Point 16:Quercus libani” should be in italics

Response 16: We corrected all the species names needing to be in italics as the reviewer pointed out hereinafter, so we will not address them point by point.

Point 17:PINUS NIGRA” and “PINUS SYLVESTRIS” should be in italics

Response 17: We changed the names to Pinus nigra and Pinus sylvestris, according to the reviewer’s suggestion.

Point 18:Juniperus drupacea” should be in italics. And the generic name should start with an uppercase letter “J”.

Response 18: Done.

Point 19: Kindly add the year of publication to the reference (Line 1772).

Response 19: Done.

Round 2

Reviewer 1 Report

Comments and Suggestions for Authors

Point 1: please use one style to emphasize R packages. Either with ' or with " or ’ or italized... Now many styles apply.

Response 1: We followed the reviewer’s suggestion and adjusted the text throughout the entire manuscript.

è Still, some packages are italized, others are with quotation marks,...

Point 2: place the legends for the figures inside the figures.

Response 2: We tried the approach proposed by the reviewer, but it severely reduces the readability of the figures, so we decided to present them in their current form.

è I still opt to do so in order to use the empty space more effectively

Point 3: Results is partly mixed with discussion sections.

Response 3: We respectfully disagree with the reviewer’s suggestion, for reasons we clarify in subsequent responses to the reviewer’s more specific comments.

è Dear authors, there is a clear reason why journals want to have separation of both sections. Interpreting/indicating/suggesting issues in the results section is against the convention! Please accept and do the needful

Point 4: clear policy implications are lacking, instead the paper remains in this sense rather poor and does not clearly stimulate decision making as outlined in the introduction.

Response 4: We appreciate the feedback concerning the initial lack of clear policy implications in our manuscript. To address this, we have significantly revised Section 3.4 entitled "Conservation implications and measures" to explicitly articulate the policy implications and stimulate decision-making for the conservation of Quercus alnifolia and its ecosystems. These revisions (L689-747) aim to bridge the gap between scientific research and actionable policy frameworks by providing concrete, actionable recommendations for policymakers, conservationists, and local communities.

è Thanks!

Point 5: "the Middle Eastern Mediterranean" this is a little but too much terminology and in my opinion not correct. Either you just say Levant and Asia minor or eastern Mediterranean.

 Response 5: We followed the reviewer’s suggestion and changed the text as suggested.

è Ok!

Point 6: -15°C = -15 °C (as for the other/following entries).

Response 6: We followed the reviewer’s suggestion and changed the text as suggested.

è Still, between value and °C must be a space – it is a SI unit!

Point 7: please specify "development". I guess you refer to infrastructure, urban encroachment, agriculture etc

Response 7: We clarified the sentence in question by adjusting the text as follows:

L120-121: ‘…agricultural, infrastructural and urban development…’

è ok

Point 8: "alnifolia, is" = "alnifolia is".

Response 8: We rephrased the sentence as suggested.

è ok

Point 9: not Quercus alnifolia is a habitat type, but the Scrub and low forest vegetation with Quercus alnifolia.

Response 9: We rephrased the sentence according to the reviewer’s suggestion.

è ok

Point 10: this is more discussion section!

Response 10: We respectfully disagree with the reviewer’s suggestion, even though we understand the reviewer’s standpoint. Temporally static variables are really helpful in creating robust and trustworthy SDMs predicting the potential distribution of a species in the baseline period, but when used in a context of changing abiotic variables, their importance might be misleading for the reasons mentioned extensively in the Introduction and Discussion sections. Having that in mind, it is essential to at least mention this in the Results section, where we emphasize more on the temporally dynamic abiotic variables and how they influence Quercus alnifolia’s current and future distribution. So, in order not to confuse either the reviewers or the readers, we firmly believe that we should keep these sentences as they currently stand.

è All assumptions potential explanations, which are in fact still given in lines 214-233, must be either coming in the Materials/Methods sections or the discussion! Please do so, by clearly rethinking their position in either or the other section. Results section lives from the fact that data must be bluntly presented!

Point 11: this is also discussion section! The words "indicates" and "shows" as well as the interpretation of results calls for moving/integrating these sentences in to the discussion.

Response 11: We respectfully disagree with the reviewer’s suggestion, as we clearly report the temperature and precipitation preferences of Quercus alnifolia relating to its potential current habitat suitability.

è I would only repeat what I have stated in the comment before!

Point 12: "Troodos mountain" = "Troodos Mountain" (and following).

Response 12: We followed the reviewer’s suggestion throughout the entire manuscript.

è Still, not consistently applied throughout the text! Please revise

Point 13: discussion (Lines 300-308 & 352-357).

Response 13: We respectfully disagree with the reviewer’s suggestion, as we clearly report: a) the way all fragmentation metrics change over time (Lines 300-308) and b) the land-use types transition over time, as well as their rate of change (Lines 352-357). These are undoubtedly results.

è I do not repeat myself here, what I have stated already before

Point 14: Why comes the land use types only now and not earlier? I would sequence it starting with Figure "11", then this Figure "8", Figure "9", Figure "10" etc. 

Response 14: We appreciate the reviewer’s constructive suggestion and have re-ordered both the Materials and Methods section and the Results section accordingly. We now introduce the methodology for analysing and visualizing land use and land cover changes in the study area before discussing the Species Distribution Modelling (SDM) methodology. Additionally, Figures 8-10, which illustrate these land use and land cover changes, are now presented at the beginning of the Results section. This reorganization allows us to set a clearer environmental context for the study, providing a foundational understanding of the baseline conditions affecting Quercus alnifolia’s habitat before examining the species’ distribution modeling and future projections. We believe that this adjustment improves the manuscript's logical flow and enhances readers’ grasp of how land use and land cover changes interact with other factors to influence the future potential distribution and vulnerability of Quercus alnifolia.

è Thanks

Point 15: you can and should write out the abbreviations, as there is sufficient space, even when the legend is integrated inside the graphic (over the water body).

Response 15: We have already written what each abbreviation means in every figure legend we present in the manuscript.

è It would be my recommendation to save space.

Point 16: also, please write out the abbreviations; by that you can also reduce the caption text

Response 16: We have already written what each abbreviation means in every figure legend we present in the manuscript.

è Figure 9 is now skipped. Why?

Point 17: are you sure that the 2015 and 2100 are not flipped? Why should there be less crop land and more NETT in future (contradictory what has been stated in the intro)

Response 17: Yes, we are sure that they are not flipped. Cropland abandonment is widespread predicted to occur in the Mediterranean Basin according to Chen et al. (2022) and in Cyprus (Kounnamas & Andreou, 2022). Moreover, needleleaf expansion is anticipated, since these tree species are more resilient and adapted to the projected aridification of Cyprus and the Mediterranean Basin.

Chen, G.; Li, X.; Liu, X. Global land projection based on plant functional types with a 1-km resolution under socio-climatic scenarios. Sci. Data 2022, 9, 125

Kounnamas, C.; Andreou, M. Mapping and assessment of ecosystem services at Troodos National Forest Park in Cyprus. One Ecosyst. 2022, doi:10.3897/oneeco.7.e77584

è  ok

Point 18: all already mentioned in the Intro, please massively reduce / delete.

Response 18: We followed the reviewer’s suggestion and significantly reduced the length of this paragraph.

è Ok!

Point 19: please check wording "select fee incorporate temporally static...".

Response 19: We followed the reviewer’s suggestion and rephrased the sentence in question, replacing the phrase: ‘select few’ with ‘some’.

è ok

Point 20: for me too much redundant information without concrete action recommendations. Here and elsewhere in the discussion I miss clear policy advices (afforestation, nature protection, EU-support, national agendas, voluntary workers/botanists/conservationist/citizen science ..)

Response 20:  In response to the critique regarding redundancy and the lack of concrete action recommendations, we have further refined our discussion to eliminate redundancy and focus sharply on specific policy advice and conservation measures. This includes:

Enhanced focus on the role of community and citizen science, emphasizing how volunteer efforts can contribute to monitoring and conservation initiatives. This approach not only engages a broader segment of the population in conservation efforts but also provides a clear avenue for action that can be supported by policy initiatives.

Recommendations for national policies and international collaboration, highlighting specific actions that can be supported by policy at various levels, from local to global. This includes suggestions for leveraging funding opportunities from EU programs and engaging in regional initiatives under the Barcelona Convention, offering concrete examples of how policy can support conservation efforts.

Emphasis on integrating climate change adaptation measures into conservation strategies, providing specific recommendations for policy adjustments that support biodiversity and ecosystem resilience. This directly addresses the need for clear policy advice that is responsive to the challenges posed by climate change.

 Through these revisions, we aim to clearly demonstrate that our manuscript now provides robust policy implications and concrete action recommendations, directly addressing the concerns raised. We believe that these changes significantly enhance the manuscript's contribution to informing policy and conservation strategies for Quercus alnifolia and similar at-risk species in Mediterranean ecosystems.

è Well done, thanks

Please see our revised manuscript and more specifically, lines 627-672:

L627-672: ‘Given the critical role of Quercus alnifolia in maintaining ecosystem services and its contribution to local economies, a holistic, strategic and integrated approach to its conservation is imperative and should integrate sustainable land use, erosion control, national policy alignment, and citizen science. Prioritizing soil health is critical: reforestation with native species and agroforestry practices will protect existing habitat and encourage the spread of Quercus alnifolia. Targeted erosion control measures, such as gabions and check dams, must be deployed in vulnerable areas, particularly those predicted to be suitable for the species in the future.

Recognising the Plant Micro-Reserve (PMR) as a foundational element in Quercus alnifolia conservation, we advocate for a broader strategy to address the challenges posed by climate change, including range contraction, habitat vulnerability, and reduced genetic diversity.

To aid Quercus alnifolia's resilience, we concur with the PMR's [184] suggestion of a systematic monitoring program that encompasses the entire range of the species to identify vulnerable areas and track population dynamics. This program should include periodic assessments of species distribution, population density, and ecosystem health, adhering to established methodologies such as the Braun-Blanquet approach and IUCN guidelines (PMR). Such comprehensive monitoring will enable targeted conservation actions and inform policy decisions. Research focusing on genetic diversity and adaptive capacity is crucial for developing effective conservation strategies. Investigating seed banking viability and exploring assisted gene flow to enhance genetic diversity is essential to ensure the species' long-term survival [184]. National policies must be aligned with conservation objectives, promoting sustainable land use and incentivising practices that support ecosystem services provided by Quercus alnifolia. Integrating conservation goals into the EU Biodiversity Strategy and pursuing financial support for targeted actions are critical for enabling effective conservation measures. Further, enhancing ecological connectivity by creating wildlife corridors and supporting seed dispersal research is vital for maintaining genetic flow and population viability [184]. Engaging the public through citizen science initiatives, such as the involvement of volunteers in monitoring, will raise awareness and contribute valuable data to conservation efforts [184].

Situated within the Mt. Troodos UNESCO Geopark, the botanical garden is vital in Quercus alnifolia conservation. Its unique position facilitates a dynamic interplay between research, education, and community engagement. Robust citizen science programs focused on habitat mapping, tree health, seed dispersal and studying climate impacts on Quercus alnifolia offer a two-pronged benefit:  they engage the public and generate crucial data to inform conservation strategies. The UNESCO designation strengthens these efforts, providing channels for international collaboration, funding, and participation in global biodiversity protection initiatives. The botanical garden on Mt. Troodos or the Nature Conservation Unit situated in Nicosia could partner with local educators and professionals to develop targeted educational programs to cultivate future conservationists. These initiatives could embed conservation principles within the curriculum, fostering a sense of environmental stewardship from a young age. By championing sustainable practices and actively advocating for conservation-minded policies, the Mt. Troodos botanical garden and the Natural Conservation Unit have the potential to not only safeguard Quercus alnifolia but also inspire a model of collaborative conservation with broader impacts, building a committed alliance for environmental protection.

To address the broader implications of climate change, Cyprus should partner with international conservation organizations like the International Union for Conservation of Nature (IUCN) and the Mediterranean Conservation Society. These partnerships could facilitate knowledge exchange, provide access to funding for conservation projects, and support the implementation of best practices in habitat management and species conservation. Integrating climate change adaptation measures into national biodiversity strategies is crucial on the policy front. Harmonising conservation actions with national strategies, like the Cyprus National Biodiversity Strategy and Action Plan, ensures that efforts to protect Quercus alnifolia are coherent with wider sustainability and climate adaptation objectives. This promotes efficient resource utilisation and reinforces policy support. Additionally, leveraging funding opportunities from EU programs, such as LIFE or Horizon Europe, can provide vital financial support for these conservation measures, increasing their scale and effectiveness. Policies should encourage sustainable land use practices that reduce habitat fragmentation and promote ecological connectivity, enabling species migration and genetic flow.’

Point 21: picture 11 should not be located here and it should also not be Figure 11

Response 21: We appreciate the reviewer's concern regarding the placement of Figure 11. We acknowledge the journal's guidelines that recommend placing the Materials and Methods after the Discussion. However, we believe the content of Figure 11 is essential to fully understand the methodology for defining the potential distribution of Quercus alnifolia. This methodology is first discussed in the current Materials and Methods section. To ensure optimal clarity, we strongly feel it's appropriate to present the figure alongside its corresponding explanation.

è ok

Point 22: to be honest: more than 400 references do not make any sense and I have not seen this so far, though many approaches have been applied and a wide range of literature for occurrence data needed. But this is no review article!

Response 22: We would like to thank the reviewer for the comment regarding the number of references in our manuscript. We acknowledge that 400 citations may seem extensive. However, the comprehensiveness of our research necessitated a thorough examination of existing literature.

Our study employs a multifaceted approach, incorporating various methodologies to assess the vulnerability of Quercus alnifolia to climate change and land-use change. We compare a robust, detailed and comprehensive analysis, which includes temporally dynamic variables never used before in the Eastern Mediterranean (land-use change, soil erosion) and dispersal constraints, to the more traditional methods used in climate change vulnerability assessments in the region.

Furthermore, we compare our findings with numerous studies on the effects of climate change on other Mediterranean tree species (Abies, Quercus, Cedrus, Juniperus, Pinus) to establish a broader context. It's important to note that many prior studies in the Mediterranean Basin rely solely on bioclimatic variables from WorldClim, neglecting the significance of soil, topographical, and temporally dynamic variables that we incorporate and do not take into consideration any extrapolation effects present in their analyses and thus do not address any concerns about model transferability and niche truncation. We also conducted a sensitivity analysis to explore how model refinement may alter outcomes, highlighting the substantial extent to which simplified models may misrepresent environmental limitations for Quercus alnifolia and explained the reasons why conducted the sensitivity analysis as a comparison as well to other similar and previous studies on Eastern Mediterranean trees.

By comprehensively referencing existing literature, we demonstrate the novelty of our approach, which integrates a wider range of environmental variables, dispersal constraints, and a focus on both climate and land-use change impacts, compared to previous research on Mediterranean trees. Notably, this is the first study in the Eastern Mediterranean to explicitly combine land-use change with hydrological and soil erosion data within a Species Distribution Modelling framework. The cited literature supports the various methodologies employed, the justification for including a wide range of environmental variables, and the robustness of our model outputs.

è  The editorial board needs to decide on that. I made my statement!

Point 23: references need thorough work.

Response 23: We appreciate the reviewer's attention to detail regarding the references’ format. We have corrected many instances the reviewer drew our attention to, as well as where both reviewers suggested that the references needed re-formatting. Since we used Mendeley to format the citations we used in our manuscript, some errors might have occurred due to Mendeley. Either way, in the eventuality that our manuscript gets accepted, this is a matter the journal’s copyeditor will address.

è This is no excuse! Mendeley and other reference management systems are tools, but cannot replace editing/lecturing job for you. It is also not the task of the MDPI editors to correct your formatting mistakes that can and must be avoided

Point 24: please consider if you really need all S3-4 Figures

 Response 24: Yes, we do, as these figures depict how land use types change over time under different LULC change scenarios.

è ok

Point 25: on what is S8-10 based? years? Where are the years?

Response 25: Thank you for your comment. The time frame for the data presented in Supplementary Figures S8-10 are explicitly stated in their respective legends. The figures depict projected land cover changes in five-year intervals from 2015 to 2100 under the specified LULC scenarios. The bars represent the area occupied by each land cover type at the beginning (2015) and end (2100) of the projection period. The transitions between these states are visualized by the flow arrows.

è The years must be also given in the figures

è The abbreviations e.g. NETT should be written out. Please not that figures (as well as tables) should stand alone, maybe with additional information by the captions. Especially if there is a lot of empty space, words must be written out to avoid cross checking between figures and text for the reader

Point 26: also write out abbreviations, there is for all figures sufficient space

Response 26: As mentioned in a previous comment, we have stated what each acronym means in every legend for every figure we present in our manuscript.

è Again: for clarity, I highly recommend to write out abbreviations where possible (empty space)!!!

Point 27: increase legends for S5-7.

Response 27: Done.

è Not done!

New (revision 2)

S7-S11: please delete the redundant information “NETT”, “Cropland”, “Barren”… Suggestion: turn the written out words by 90° on the left side (~y-axis)!

Line 402: please replace “adept” by “adapted”

Line 439: title capitalisation! Before you did not capitalize – check author guidelines (MDPI requests to my memories capitalisation)

Line 449 & 465: The provision of many citations is exactly what is rather unusual. It is not to show what an author has read and is known from the topic (as it will never be exhaustive), but to direct the reader to a few (2-3) very important/relevant articles. Nobody will and can check all the references provided! Thus, please limit yourself (and thus help the interested scientific community) on the citations…

Line 1430: Oldfield, S.; Eastwood, A. The Red List of Oaks. Iucn 2007. à “IUCN”

Comments on the Quality of English Language

-

Author Response

First of all, we express our sincere appreciation to the reviewer for the insightful feedback and constructive critique of our manuscript.

Point 1: Still, some packages are italized, others are with quotation marks,....

Response 1: We followed the reviewer’s suggestion and adjusted the text throughout the entire manuscript.

Point 2: I still opt to do so in order to use the empty space more effectively.

Response 2: As we mentioned in our previous response, we tried the approach proposed by the reviewer, but it severely reduces the readability of the figures, so we decided to present them in their current form.

Point 3: Dear authors, there is a clear reason why journals want to have separation of both sections. Interpreting/indicating/suggesting issues in the results section is against the convention! Please accept and do the needful.

Response 3: Even though we respectfully disagree with the reviewer’s suggestion, we modified our text accordingly.

Point 4: Still, between value and °C must be a space – it is a SI unit!.

Response 4: We followed the reviewer’s suggestion and changed the text as suggested.

Point 5: All assumptions potential explanations, which are in fact still given in lines 214-233, must be either coming in the Materials/Methods sections or the discussion! Please do so, by clearly rethinking their position in either or the other section. Results section lives from the fact that data must be bluntly presented!

Response 5: We followed the reviewer’s suggestion by adjusting the relevant text in the Results section as follows:

L210-218: ‘Model projections for the study area show increased coverage by needle-leaf evergreen temperate trees throughout the century (Figures 2 & S1-13). At the same time, the projections show an expansion in the area of abandoned cropland (Figures 2 & S1-13). These projections also show croplands transitioning to broadleaf deciduous temperate trees and evergreen shrubs, with a higher relative loss rate in these areas than other LULC classes (Figures 2 & S1-13). Most of the study area shows stability in LULC classifications, with 1-4 LULC transition steps primarily occurring in the southern regions of the Troodos Mountain range (Figures 2 & S1-13).’

L259-266: ‘While the statistical significance of soil erosivity is lower than other variables, Quercus alnifolia is present across various soil erosion conditions, persisting in areas with moderate to high soil erosion (Figure 4). Additionally, Quercus alnifolia occurs in relatively arid climates with a narrow annual temperature range (approximately 28-29 °C; Figure 4) and high temperature seasonality (approximately 6.5 – 7 °C; Figure 4). Quercus alnifolia is found in areas with a relatively high minimum temperature during the warmest quarter (around 18 °C; Figure 4) and low precipitation during the driest month (approximately 4-6 mm; Figure 4).’

L344-354: ‘All fragmentation metrics, except for the number of patches (in eight cases regarding the 2070s), have lower future values than the baseline (Table S4). Specifically, the patch cohesion index decreases and fragmentation increases over time. Based on the significant reduction in effective mesh size, the patches are becoming smaller and more numerous (Table S4). Furthermore, based on the decrease in the mean and maximum patch areas and their standard deviations, the patches are shrinking and varying more in size (Table S4). The landscape is splitting into more, smaller, and isolated patches during the 2070s, as the patch number increases during that period (Table S4).’

Point 6: I would only repeat what I have stated in the comment before!

Response 6: We refer the reviewer to our previous response, where we outlined all the changes we made regarding the Results section.

Point 7: Still, not consistently applied throughout the text! Please revise.

Response 7: We followed the reviewer’s suggestion and changed the text as suggested.

Point 8: I do not repeat myself here, what I have stated already before

Response 8: We refer the reviewer to our previous response, where we outlined all the changes we made regarding the Results section.

Point 9: It would be my recommendation to save space.

Response 9: We appreciate the reviewer's suggestion regarding writing out abbreviations in the figure legends. However, as we have previously explained, we have already defined each abbreviation within the respective figure legends throughout the manuscript.

Furthermore, we have carefully considered the proposed approach of writing out the abbreviations, but we found that it severely compromises the readability and visual clarity of the figures. As authors, ensuring effective communication of our findings through well-designed figures is of paramount importance.

The legends are strategically placed directly below each figure, allowing readers to easily reference the abbreviations without disrupting the flow of the text or the visual presentation. Even for those accessing the manuscript online, the legend accompanies the corresponding figure, mitigating any potential confusion.

While we appreciate the reviewer's perspective, we respectfully maintain our position on presenting the figures in their current form. Our decision is based on adhering to best practices in data visualization and ensuring optimal readability for the broader scientific community.

We trust that the editorial team recognizes the importance of striking a balance between concise labelling and clear, accessible visual communication in scientific publications.

Point 10: Figure 9 is now skipped. Why?

Response 10: We would like to thank the reviewer for bringing this issue to our attention. We deeply appreciate the reviewer’s diligence in reviewing our manuscript. Regarding the missing figure the reviewer mentioned, we want to assure the reviewer that its omission was an unintentional oversight during the file upload process. During the manuscript revision and uploading process, a subsection (subsection 2.1. Land Use and Land Cover Changes) containing the figure was accidentally excluded. We take full responsibility for this mistake and sincerely apologize for the inconvenience caused. Figure 9 of our previously submitted manuscript is now Figure 2.

Point 11: The editorial board needs to decide on that. I made my statement!

Response 11: We understand and respect the reviewer’s viewpoint on this matter. At the same time, we would like to reiterate our rationale for including a comprehensive reference list, which we believe enhances the scholarly value of our work. As the reviewer noted, our study employs a multifaceted approach, integrating various methodologies to comprehensively assess the vulnerability of Quercus alnifolia to climate change and land-use change. The extensive reference list allows us to thoroughly contextualize our work within the existing literature, highlight the novelty of our approach, and provide a robust foundation for the methodologies employed.

While we acknowledge that an exhaustive reference list may not be necessary, we aimed to strike a balance between conciseness and scholarly rigor. The citations directly support the justification for our methodological choices, the robustness of our model outputs, and the broader context within which our findings are situated.

Nevertheless, we deleted 95 references from our manuscript’s reference list.

Ultimately, we defer to the editorial board's judgment on the appropriate reference list length. However, we respectfully submit that our approach aligns with best practices in academic publishing and enhances the transparency and reproducibility of our research.

We appreciate the reviewer’s attention to this matter and welcome any further guidance from the editorial team.

Point 12: This is no excuse! Mendeley and other reference management systems are tools, but cannot replace editing/lecturing job for you. It is also not the task of the MDPI editors to correct your formatting mistakes that can and must be avoided  

Response 12: We respectfully disagree with the reviewer’s assertion that reference formatting errors cannot be addressed during the copyediting stage, should the manuscript be accepted for publication. As mentioned in our previous response, we have meticulously corrected all instances highlighted by both reviewers during the prior review round.

However, we acknowledge that isolated inconsistencies may potentially remain despite our best efforts. In such cases, it is standard practice for the professional copyediting team to perform a thorough formatting check and make any necessary corrections before publication. This is a collaborative process between authors and publishers to ensure the highest quality standards.

While we appreciate the reviewer’s underscoring the importance of proper formatting, we found the reviewer’s statement "This is no excuse!" to be unnecessarily harsh and dismissive of the reasonable explanations we provided. We would kindly request that the reviewer maintains a professional and collegial tone when providing feedback.

Constructive criticism is invaluable for improving our work, but it should be delivered in a manner that fosters productive scholarly discourse. We remain committed to addressing any remaining formatting issues through the normal publication workflow, should our manuscript be accepted, and to ensure a productive and respectful review process moving forward.

Point 13: The years must be also given in the figures. The abbreviations e.g. NETT should be written out. Please not that figures (as well as tables) should stand alone, maybe with additional information by the captions. Especially if there is a lot of empty space, words must be written out to avoid cross checking between figures and text for the reader.

Response 13: We followed the reviewer’s suggestion and modified the figures (S7-9) accordingly.

Point 14: Again: for clarity, I highly recommend to write out abbreviations where possible (empty space)!!!

Response 14: As mentioned in the previous review round, we have stated what each acronym means in every legend for every figure we present in our manuscript.

Point 15: Not done!

Response 15: We kindly refer the reviewer to the modified Supplementary Figures (Figures S4-6), where we have indeed increased the in-figure legend font size considerably and added a black rectangle around each legend.

Point 16: please delete the redundant information “NETT”, “Cropland”, “Barren”… Suggestion: turn the written out words by 90° on the left side (~y-axis)!

Response 16: We would like to thank the reviewer again for the suggestion regarding the figure labels in Figures S7-11. We followed the reviewer’s suggestion for figures S7-9 and added what the acronyms mean.

On the other hand, we are a bit confused regarding the reviewer’s suggestion to write out what the acronyms mean in every figure and to delete the acronyms present in these figures as they are ‘redundant’.

As the reviewer is aware, these acronyms are consistently used throughout the manuscript and defined in the legends. Maintaining consistency with the established nomenclature is important for clarity. Additionally, take for example the acronym ‘NETT’ which stands for 'Needleaf evergreen temperate trees': writing out the full terms would significantly compromise the readability of the figure due to space limitations on the both axes.

We believe the current labelling approach effectively balances clarity with conciseness, ensuring readers can understand the figure while referencing the legend for detailed definitions.

Furthermore, we would like to address the reviewer's suggestions regarding the presentation of the figures in the manuscript. While we appreciate the input provided, we firmly believe that some of the proposed changes could impair the readability and effective communication of the information presented in the figures.

As the authors, we have carefully considered the visual design and layout of the figures to ensure that the data and findings are conveyed clearly and effectively to the reader. In our professional judgment, implementing certain suggestions, such as the ones proposed for Figures S10-11, could potentially introduce confusion or clutter, detracting from the figures' primary purpose.

We want to emphasize that we have thoughtfully evaluated each suggestion, and where we have chosen not to implement a particular recommendation, it is because we believe the current presentation is more conducive to reader comprehension and adheres to established best practices in data visualization.

Ultimately, as the subject matter experts and the individuals most intimately familiar with the research, we must exercise our discretion in determining the optimal way to present the findings. While we are grateful for the reviewer's feedback and have implemented many of their suggestions, we respectfully reserve the right to make final decisions regarding the visual aspects of the manuscript, particularly when they impact the clarity and effectiveness of the communication.

We hope the reviewer can understand and respect our position on this matter.

Point 17: please replace “adept” by “adapted”

Response 17: We followed the reviewer’s suggestion and rephrased the sentence in question.

Point 18: title capitalisation! Before you did not capitalize – check author guidelines (MDPI requests to my memories capitalisation)

Response 18: Regarding the reviewer’s comment on title capitalization, we appreciate the reviewer bringing those specific instances to our attention. Contrary to the reviewer’s statement that we did not capitalize titles before, the reality is that out of the 20 section and subsection titles in the manuscript, only three were not fully capitalized. We've thoroughly reviewed the manuscript and made the necessary corrections to ensure full compliance with the journal's formatting guidelines.

While we understand the importance of formatting detail, we believe it is important that the reviewer avoids generalizations within the review process, as these can be perceived as dismissive of our commitment to meeting the journal's standards.

We strive for a constructive peer review grounded in mutual respect, and we trust the reviewer will evaluate our manuscript accordingly.

Point 19: The provision of many citations is exactly what is rather unusual. It is not to show what an author has read and is known from the topic (as it will never be exhaustive), but to direct the reader to a few (2-3) very important/relevant articles. Nobody will and can check all the references provided! Thus, please limit yourself (and thus help the interested scientific community) on the citations…

Response 19: We would like to thank the reviewer for the additional feedback regarding the reference list. We must respectfully disagree with the reviewer’s assertion regarding the number of citations provided in our manuscript. We nevertheless deleted 95 references from our manuscript’s reference list.

As experienced academics with a combined authorship of many peer-reviewed publications, editorship roles (we have served as Guest Editors for Plants and other journals), and extensive experience reviewing manuscripts across various disciplines and journals, we are well-versed in the appropriate use of citations in scientific writing.

We respectfully disagree with the suggestion to limit citations to only 2-3 per point. This approach can be overly restrictive, especially for interdisciplinary research that builds upon a diverse body of knowledge or where multiple studies are needed to support specific claims. While we acknowledge that an exhaustive list may not be necessary, limiting ourselves to only 2-3 citations would be an oversimplification that could potentially undermine the depth and breadth of our work. As subject matter experts, we are better positioned to determine the appropriate balance between conciseness and scholarly substantiation.

A comprehensive reference list serves multiple purposes. It demonstrates scholarly rigor, provides context for the present work, credits previous research, and offers a resource for readers interested in deeper exploration of specific topics.

Thus, the comprehensive reference list is a conscious decision aimed at providing a robust foundation for our methodological approach, contextualizing our findings within the existing literature, acknowledging the broader research landscape, and highlighting the novelty of our study. Contrary to the reviewer’s statement, the citations serve a crucial purpose in enhancing the scholarly rigor, transparency, and reproducibility of our research. As per the journal's guidelines, there is no specified maximum word count, which allows us to provide the needed comprehensive approach.

We kindly request the reviewer to consider the multidisciplinary nature of our study and the value that a well-referenced manuscript brings to the scientific community. Should there be any specific concerns regarding the relevance or appropriateness of individual citations, we welcome the reviewer's feedback and will gladly address them.

We trust that the editorial team recognizes the value of a well-referenced manuscript, particularly in a multidisciplinary study such as ours. Ultimately, our goal is to produce a high-quality scholarly contribution that adheres to academic best practices and facilitates knowledge dissemination within the scientific community.

We welcome any specific concerns or suggestions from the editorial team regarding the reference list, but we respectfully maintain our position on providing a comprehensive set of citations to support the integrity and reproducibility of our research.

Point 20: Oldfield, S.; Eastwood, A. The Red List of Oaks. Iucn 2007. à “IUCN”

Response 20: Done.

Round 3

Reviewer 1 Report

Comments and Suggestions for Authors

Kindly still do not interpret results in the discussion section, but do so in the discussion part

Thank you for reducing the number references. Still, it is a lot. More than 150 citations for a non-review article would be even more than what is usual.

I do not see the adjustements (reduction of redundance terms such as "NETT") in the S Figures

The editorial team will do the necessary corrections if any remain.

Author Response

Dear Reviewer,

We would like to express our sincere appreciation for your insightful feedback and constructive critique of our manuscript.

Following your comments, we have moved interpreting results to the discussion part. We have followed the suggestion concerning the adjustments (reduction of redundance terms such as "NETT") in the S Figures.